# The Mirage of Performance Gains:
# Why Contrastive Decoding Fails to Mitigate Object Hallucinations in MLLMs?

**Hao Yin**[1]    **Guangzong Si**[1,2]    **Zilei Wang**[1,*]

[1] University of Science and Technology of China
[2] Eastern Institute of Technology, Ningbo
{yinhnavi, guangzongsi}@mail.ustc.edu.cn, zlwang@ustc.edu.cn

## Abstract

Contrastive decoding strategies are widely used to reduce object hallucinations in multimodal large language models (MLLMs). These methods work by constructing contrastive samples to induce hallucinations and then suppressing them in the output distribution. However, this paper demonstrates that such approaches fail to effectively mitigate the hallucination problem. The performance improvements observed on POPE Benchmark are largely driven by two misleading factors: (1) crude, unidirectional adjustments to the model's output distribution and (2) the adaptive plausibility constraint, which reduces the sampling strategy to greedy search. To further illustrate these issues, we introduce a series of spurious improvement methods and evaluate their performance against contrastive decoding techniques. Experimental results reveal that the observed performance gains in contrastive decoding are entirely unrelated to its intended goal of mitigating hallucinations. Our findings challenge common assumptions about the effectiveness of contrastive decoding strategies and pave the way for developing genuinely effective solutions to hallucinations in MLLMs. The source code is available at https://github.com/ustc-hyin/cd_rethink

## 1   Introduction

The hallucination problem [1, 2, 3, 4] in multimodal large language models (MLLMs) [5, 6, 7] refers to the generation of outputs that are factually incorrect or misaligned with the input data. This issue arises from challenges in aligning diverse data modalities, such as text and images, which amplify reasoning errors. Such hallucinations can have serious consequences in critical domains, including autonomous driving [8, 9, 10, 11] (e.g., false object detection leading to accidents) and healthcare [12, 13] (e.g., incorrect diagnostic interpretations).

Contrastive decoding methods [14, 15, 16] are widely recognized as an effective approach to addressing object hallucination in generative models. As illustrated in Figure 1, these methods construct contrastive samples designed to induce hallucinations, then suppress the corresponding output distributions, ensuring closer alignment between model outputs and visual inputs. Representative approaches within this framework include Visual Contrastive Decoding (VCD) [17], Instruction Contrastive Decoding (ICD) [18], and Self-Introspective Decoding (SID) [19]. Their training-free nature and purported ability to address hallucinations have made them highly regarded in the field.

Although methods like VCD have demonstrated remarkable performance improvements on the POPE benchmark [1], we reveal in Section 4 that these results are highly misleading. In reality, these

---

[*]Corresponding Author

39th Conference on Neural Information Processing Systems (NeurIPS 2025).

methods fail to effectively address model hallucination. The observed performance gains on the POPE benchmark are primarily driven by two factors:

> **Misleading Nature of Performance Improvement**
>
> $\mathcal{R}_1$: *A unidirectional adjustment of the output distribution, which simply biases the model towards producing more "Yes" outputs, leading to a balanced distribution on certain datasets.*
>
> $\mathcal{R}_2$: *The adaptive constraints in these methods degrade the sampling decoding strategy into an approximation of greedy search, resulting in deceptively improved performance.*

To expose the misleading nature of the improvement in the first scenario, we implemented two forced distribution adjustment algorithms in Section 5.1 to show that the apparent gains of contrastive decoding on the POPE Benchmark are not genuine. The methods are as follows: (1) Prompt-Based Adjustment, where we added a prompt to the instruction, such as *"Whenever possible, please select Yes."* to bias outputs toward "*Yes*"; and (2) Output Layer Modification, where we altered the output layer to favor "*Yes*" when the probabilities for "*Yes*" and "*No*" were similar. Although neither method mitigates hallucinations, both achieved performance gains comparable to those of contrastive decoding, confirming that these improvements do not represent a genuine solution to the problem.

To highlight the misleading nature of the performance improvement in the second scenario, we incorporated the adaptive plausibility constraint into the standard sampling strategy and compared its predictions with those from contrastive decoding in Section 5.2. The experimental results reveal that, despite having no theoretical connection to hallucination mitigation, the adaptive plausibility constraint accounts for nearly all the performance gains attributed to contrastive decoding. This finding underscores that the contrastive decoding methods, in essence, fail to mitigate hallucinations.

Overall, this paper makes the following three contributions:

- We identified that the performance improvement of contrastive decoding methods stems from its unidirectional and blunt adjustment of the output distribution, which coincidentally balances the distribution on certain datasets.
- We discovered that another key factor driving the performance gains of contrastive decoding methods is their adaptive plausibility constraints, which streamline the sampling strategy into an approximation of greedy search.
- We developed a series of spurious improvement methods and evaluated their performance against contrastive decoding methods. Our findings convincingly show that contrastive decoding methods do not alleviate hallucinations in any meaningful way.

## 2   Related Work

**Multimodal Large Language Models.** The evolution of MLLMs [20, 21] has progressed from BERT-based decoders [22, 23] to advanced LLM architectures [24, 25], enabling more effective multimodal relationship modeling [26, 27]. Models such as BLIP-2 [28] and MiniGPT-4 [29] employ Q-Former mechanisms to enhance the alignment between visual and textual inputs, facilitating more precise cross-modal interactions. InstructBLIP [30] extends this framework by integrating task-specific instructions, improving the model's ability to interpret context-sensitive visual semantics. Meanwhile, LLaVA [31, 32] and Qwen-VL [33] adopt simpler linear projection methods that streamline alignment, leading to superior performance in vision-language tasks. Despite these advancements, hallucination remains a persistent challenge that warrants further investigation.

**Contrastive Decoding Strategies.** Contrastive decoding [34, 35, 36] are widely recognized as effective in addressing object hallucination in generative models. Visual Contrastive Decoding (VCD) [17] addresses object hallucination by comparing output distributions generated from standard visual inputs and distorted visual inputs. This approach reduces the model's dependence on linguistic priors within integrated LLMs and minimizes the impact of statistical biases in MLLM pretraining corpus. Instruction Contrastive Decoding (ICD) [18], in contrast, focuses on the role of instruction perturbations in amplifying hallucinations. By examining the differences in output distributions between standard and perturbed instructions, ICD detects hallucination-prone content and mitigates its impact effectively. Building upon these two hallucination mitigation methods, numerous approaches, including Adaptive Focal-Contrast Decoding (HALC) [37], Self-Introspective Decoding (SID) [19],

and Visual Layer Fusion Contrastive Decoding (VaLiD) [38], have been developed based on similar principles. Although these methods have demonstrated substantial performance improvements on the POPE Benchmark, we will show that these improvements are, in fact, **entirely unrelated to the original objective of hallucination mitigation**.

## 3 Preliminary

This section outlines the main components and operational workflows of three leading hallucination mitigation methods: VCD, ICD, and SID, as illustrated in Figure 1. All three adopt contrastive decoding strategies to reduce hallucinations and improve consistency with the visual input. The POPE Benchmark is also discussed as one of the most important metrics for evaluating their performance. Further technical details of VCD, ICD, and SID can be found in Section A.

***Vanilla Decoding.*** We consider a MLLM parametrized by $\theta$. The model takes as input a textual query $x$ and a visual input $v$, where $v$ provides contextual visual information to assist the model in generating a relevant response $y$ to the textual query. The response $y$ is sampled auto-regressively from the probability distribution conditioned on the query $x$ and the visual context $v$. Mathematically, this can be formulated as:

$$y_t \sim p_\theta \left( y_t \mid v, x, y_{<t} \right) \propto \exp \left( \mathrm{logit}_\theta \left( y_t \mid v, x, y_{<t} \right) \right) \tag{1}$$

where $y_t$ denotes the token at time step $t$, and $y_{<t}$ represents the sequence of generated tokens up to the time step $(t-1)$.

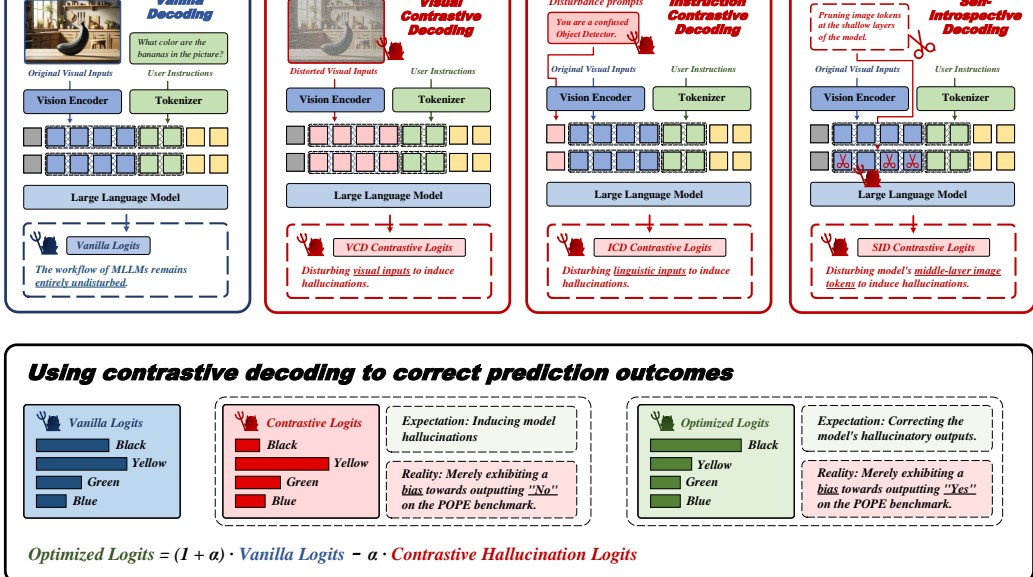

Figure 1: An illustration of hallucination mitigation methods: Visual Contrastive Decoding, Instruction Contrastive Decoding, and Self-Introspective Decoding. The hallucination induction module shifts outputs toward negative responses, while the contrastive decoding module shifts them toward positive responses, rather than achieving their intended effects.

***Visual Contrastive Decoding.*** VCD is a corrective strategy aimed at reducing hallucinations in MLLMs. Given a textual query and its corresponding image, the model produces two output distributions: one from the original image and another from a perturbed variant, typically generated by applying controlled modifications such as Gaussian noise. By evaluating the divergence between these distributions, VCD constructs a contrastive output distribution that improves the reliability and factual accuracy of the model's response.

***Instruction Contrastive Decoding.*** ICD addresses hallucinations by leveraging the observation that instruction perturbations, particularly those involving negative prefixes, increase uncertainty in multimodal alignment. ICD operates by first amplifying the probabilities of hallucinated concepts,

then systematically detaching these from the original output distribution. This contrastive process reduces the model's vulnerability to object hallucinations.

***Self-Introspective Decoding.*** SID extends the ideas of VCD and ICD by addressing a key limitation: perturbing the full input can inject too much noise, making it harder to produce useful hallucinations. To resolve this, SID adaptively prunes the input by keeping only a small set of image tokens with low attention scores after the early decoder layers (see Figure 1, far right). This focused adjustment encourages vision-and-text hallucinations to emerge during decoding. These hallucinated elements are then separated from the original output distribution to improve the model's reliability.

***Adaptive Plausibility Constraint.*** One key challenge inherent in the three aforementioned methods is the risk of indiscriminate penalization across the entire output space, which can unintentionally suppress valid predictions and, paradoxically, favor the generation of implausible tokens. To mitigate this, all three methods incorporate an adaptive plausibility constraint. This constraint dynamically adjusts penalization based on confidence scores derived from the model's output distribution, conditioned on the original visual input $v$. Formally, the constraint is defined as:

$$\mathcal{V}_{\text{head}}\left(y_{<t}\right) = \left\{ y_t \in \mathcal{V} \mid p_\theta\left(y_t \mid v, x, y_{<t}\right) \geq \beta \max_w p_\theta\left(w \mid v, x, y_{<t}\right) \right\},$$

$$p_{cd}\left(y_t \mid v, x\right) = 0 \quad \text{if } y_t \notin \mathcal{V}_{\text{head}}\left(y_{<t}\right)$$

(2)

Here, $\mathcal{V}$ represents the output vocabulary of the multimodal large language model (MLLM), and $\beta$ is a hyperparameter controlling the truncation threshold of the next-token distribution. A higher value of $\beta$ results in more aggressive truncation, thereby retaining only the most probable tokens.

***Polling-based Object Probing Evaluation.*** POPE [1, 39] offers a robust framework for evaluating object hallucinations in multimodal large language models (MLLMs). Departing from caption-based methods, it frames hallucination detection as a binary task using direct ***Yes-or-No*** questions (e.g., "Is there a chair in the image?"), enabling clearer interpretation. The benchmark ensures a balanced distribution of "*Yes*" and "*No*" samples (50% each). Contrastive decoding–based mitigation strategies have demonstrated their effectiveness primarily through improved performance on POPE, reinforcing their credibility within the research community.

## 4    Misleading Performance Improvement

In this section, we highlight two misleading factors contributing to the performance improvement of contrastive decoding methods on the POPE Benchmark: (1) *Unidirectional output adjustment skews the model towards generating more "Yes" outputs*, leading to a balanced distribution in certain datasets. (2) *The adaptive plausibility constraint degrades sampling decoding strategy into greedy search*, resulting in deceptively improved outcomes.

Table 1: Performance of various contrastive decoding methods on subsets of POPE Benchmark.

| Dataset | COCO Random | | GQA Adversarial | |
|---|---|---|---|---|
| **Method** | **Acc %** | **Yes %** | **Acc %** | **Yes %** |
| Greedy | 87.1 | 39.2 | 80.9 | 54.0 |
| VCD | **88.6** | 46.4 | **78.0** | 63.3 |
| SID | **87.9** | 42.3 | **79.9** | 57.8 |

Table 2: Output distribution generated from contrastive inputs in contrastive decoding methods.

| Dataset | COCO Random | | GQA Adversarial | |
|---|---|---|---|---|
| **Method** | **Acc %** | **Yes %** | **Acc %** | **Yes %** |
| Greedy | 87.1 | 39.2 | 80.9 | 54.0 |
| VCD-C | 76.7 | **28.2** | 71.5 | **41.3** |
| SID-C | **79.0** | **23.6** | 74.2 | **43.1** |

### 4.1    Unidirectional Output Adjustment

In this subsection, we illustrate how contrastive decoding algorithms can deceptively enhance the performance of MLLMs on the POPE Benchmark by applying targeted, unidirectional modifications to the output distribution. We begin by evaluating the performance of various contrastive decoding methods on the MSCOCO [40] and GQA [41] datasets, analyzing both accuracy and the distribution of the model's responses. For this study, we selected LLaVA-v1.5-7B as the foundational MLLM, using a greedy search decoding strategy.

As shown in Table 1, both VCD and SID significantly skewed the model's output distribution toward "*Yes*" across all subsets. On MSCOCO-Random subset, where the original output distribution was

skewed toward "*No*," VCD and SID corrected this imbalance, resulting in a more balanced distribution and improved accuracy. Conversely, for GQA-Adversarial subset, where the output distribution was already biased toward "*Yes*," these methods intensified the skew, ultimately reducing accuracy.

We further illustrate how model outputs change after applying contrastive decoding methods, providing a clearer understanding of their performance improvements. As shown in Figure 2, the method primarily alters predictions from "*No*" to "*Yes*," significantly outpacing the reverse. On the MSCOCO-Random dataset, where the output distribution is initially skewed toward "*No*," this adjustment converts many false negatives into true positives, thereby improving accuracy. Conversely, on the GQA-Adversarial dataset, which is biased toward "*Yes*," these modifications lead to the misclassification of numerous true negatives as false positives, resulting in a performance decline. For more details on prediction shifts, please refer to Section B.

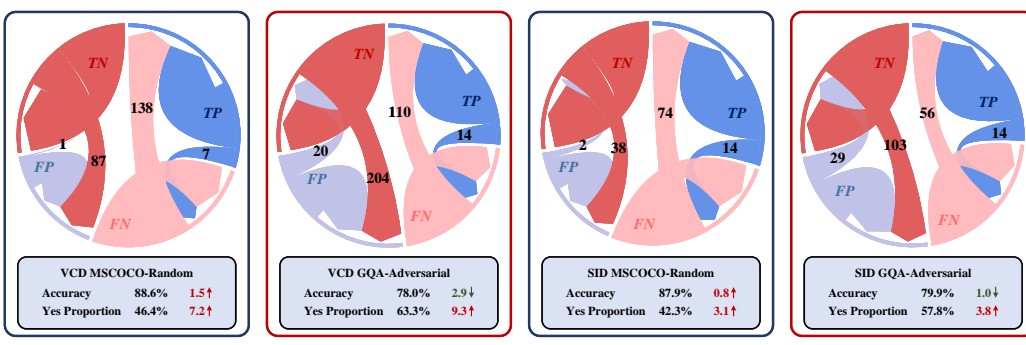

Figure 2: Changes in the distribution of predictions after applying contrastive decoding methods.

To understand why contrastive decoding methods consistently increase the likelihood of a "*Yes*" response, we analyzed the output distribution generated from contrastive samples, as shown in Table 2. In Section 3, we proposed that the primary function of contrastive samples is to induce hallucinations, allowing contrastive decoding to subsequently filter out these hallucinated elements from the output distribution. However, the results in Table 2 reveal that this objective was entirely unmet. Most outputs derived from contrastive samples were incorrect, not due to successfully induced hallucinations, but because the model overwhelmingly favored "*No*" responses. This severe bias in the output distribution led to a significant decline in accuracy. A more detailed explanation of why outputs generated from contrastive inputs are biased toward "*No*" is provided in Section C.

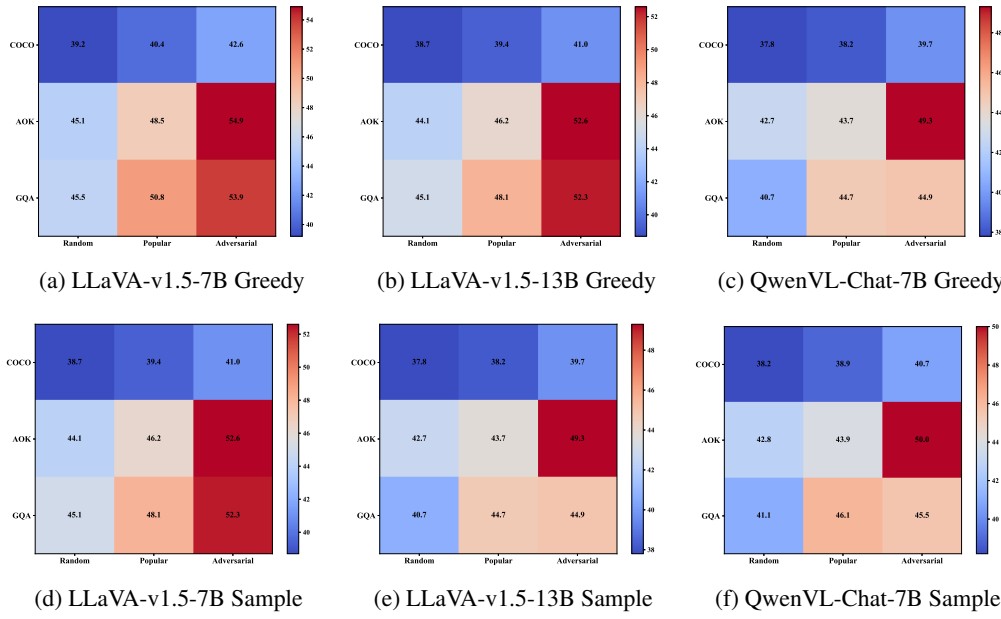

Figure 3: Skewness in the Raw Output Distribution of MLLMs across Different Datasets

Based on the above discussion, the practical performance of contrastive decoding methods is illustrated in the lower section of Figure 1. The output distribution derived from the contrastive inputs is heavily biased toward "*No*." However, the contrastive decoding method suppresses this content in the original output, thereby unilaterally increasing the model's likelihood of answering "*Yes*." Whether the model's performance on the dataset improves depends heavily on whether this increased "*Yes*" frequency leads to a more balanced output distribution. However, as shown in Figure 3, the output distribution of MLLMs tends to be biased toward "*No*" in most data subsets. Therefore, contrastive decoding methods still manage to achieve a strong overall performance on the POPE Benchmark.

## 4.2   Sampling Decoding Degradation

In this subsection, we will illustrate how contrastive decoding methods misleadingly enhance model performance by degrading sampling-based decoding strategies into greedy search through the adaptive plausibility constraint.

Notably, the POPE Benchmark, which requires models to answer "**Yes**" or "**No**," functions as a binary classification task. Consequently, greedy search is the most suitable decoding strategy, rendering sampling-based methods unjustifiable. As shown in the upper-left corner of Figure 4, experimental results further confirm that greedy search significantly outperforms direct sampling. In fact, greedy search outperforms direct sampling on the vast majority of tasks[42]. Additional comparisons of decoding strategies and their effects on prediction outcomes are provided in Section D. However, many contrastive decoding methods report performance improvements using sampling strategies, necessitating a closer examination of these claims.

We revisit the **adaptive plausibility constraint** introduced in Section 3 and formally defined in Equation (2). This constraint ensures that when the model exhibits high confidence in its outputs corresponding to the original input, the candidate pool is refined to retain only high-probability tokens. By incorporating this mechanism into contrastive decoding methods, it aims to mitigate adverse effects by preventing the generation of implausible tokens, while safeguarding the coherence and quality of the generated content.

In its original design, the constraint was intended as a complement to contrastive decoding strategies, with **no explicit connection to mitigating hallucinations**. Consequently, it was assumed to have no significant effect when applied independently. However, our findings challenge this assumption: under a sampling strategy, the constraint emerges as a **pivotal** contributor to performance gains.

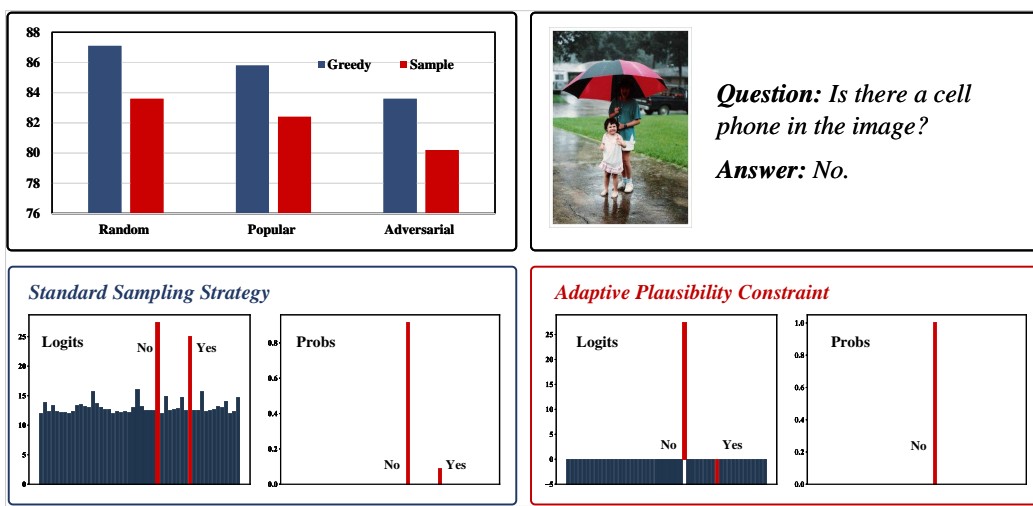

Figure 4: Why does the adaptive plausibility constraint alone result in improvements?

Figure 4 provides a detailed explanation of why the adaptive plausibility constraint alone significantly improves performance when using the sampling strategy. For the question "Is there a cell phone in the image?", LLaVA-v1.5-7B model generates the correct output distribution: *Yes*: 8.8% and *No*: 91.2%. Using a greedy search strategy, the model consistently produces the correct answer, "*No*."

However, when employing a sampling strategy, there is an 8.8% chance that the model generates the incorrect answer, "*Yes*."

When the adaptive plausibility constraint is applied, many candidate options are eliminated by setting their logits to negative infinity for failing to satisfy the condition:

$$p_\theta \left( y_t \mid v, x, y_{<t} \right) \geq \beta \max_w p_\theta \left( w \mid v, x, y_{<t} \right). \tag{3}$$

Among the excluded candidates is the option "*Yes*." Consequently, the sampling strategy reduces to a greedy search, ensuring a 100% probability of correctly answering "*No*."

Consequently, the adaptive plausibility constraint greatly limits the pool of candidate options, transforming the sampling strategy into a predominantly greedy search. As demonstrated in Figure 4, MLLMs exhibit markedly superior performance on the POPE Benchmark under a greedy strategy, underscoring the constraint's pivotal contribution to performance gains.

### 4.3 Insights

In Section 4.1, we show that when using greedy search as the decoding strategy, contrastive decoding methods modify the model's predictions in a unidirectional manner, shifting the output distribution toward *Yes*. As a result, performance improvements primarily depend on whether the model's original output distribution was biased toward *No*.

In Section 4.2, we demonstrate that when direct sampling is used as the decoding strategy, the adaptive plausibility constraint effectively reduces it to greedy search, serving as a key driver of the observed performance gains.

These findings suggest that the reported improvements from contrastive decoding may be misleading. Specifically, the gains observed on the POPE Benchmark could falsely imply effective hallucination mitigation when, in reality, they stem from unrelated factors.

## 5 Spurious Improvement Methods

In this section, we propose a series of spurious improvement methods based on the two fundamental reasons for performance improvement discussed in Section 4.3. Although these methods are entirely unrelated to hallucination mitigation, they yield experimental results comparable to contrastive decoding techniques. This evidence suggests that while contrastive decoding enhances performance on POPE Benchmark, it does not address hallucinations.

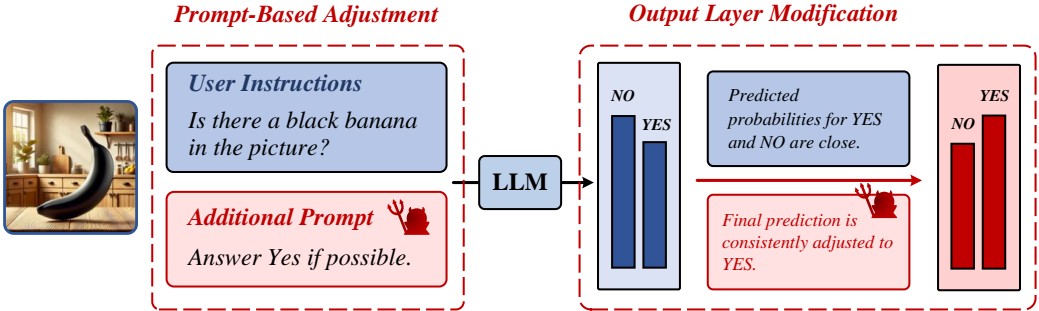

Figure 5: Illustration of Prompt-Based Adjustment and Output Layer Modification Algorithms.

### 5.1 Forced Distribution Adjustment

For the first misleading factor in performance improvement, which involves modifying model predictions in a single direction to bias the output distribution toward "*Yes*," we introduce two pseudo-performance enhancement methods: Prompt-Based Adjustment and Output Layer Modification. The implementation of these algorithms is detailed in Figure 5.

**Prompt-Based Adjustment** modifies the input side of the model by appending an additional prompt, "Answer Yes if possible," after the user's instruction. This extra input biases the model's output distribution toward "*Yes*." **Output Layer Modification** refers to adjustments made at the output stage of the model. After generating its initial prediction, the model evaluates the probabilities of "*Yes*" and "*No*." If their difference is small, i.e.,

$$|p_\theta(\text{Yes} \mid v, x) - p_\theta(\text{No} \mid v, x)| < \tau, \tag{4}$$

the prediction is forcibly set to "*Yes*." Here, $\tau$ controls how close the probabilities must be to trigger this modification. This adjustment increases the likelihood of the model predicting "*Yes*."

**Experimental Settings.** We selected QwenVL-7B, LLaVA-v1.5-7B, and LLaVA-v1.5-13B as the backbone MLLMs. For decoding, we employed a greedy search strategy. Regarding the critical POPE benchmark, our experiments were conducted on COCO dataset, where the raw output distribution of MLLMs tends to be biased toward "*No*." As a result, all contrastive decoding methods exhibited notable improvements on this dataset. In addition, we conducted experiments on several other benchmarks that have not been evaluated by many contrastive decoding methods, including MME[43], CHAIR[44], NoCaps[45] and LLaVA-Bench[31], covering both discriminative and generative tasks.

Table 3: Performance of Prompt-Based Adjustment (PBA) and Output Layer Modification (OLM).

| Category | Method | LLaVA-v1.5-7B | | LLaVA-v1.5-13B | | QwenVL-Chat-7B | |
| | | Accuracy | Yes (%) | Accuracy | Yes (%) | Accuracy | Yes (%) |
|---|---|---|---|---|---|---|---|
| Random | Greedy | 87.1 ↑0.0 | 39.2 ↑0.0 | 86.7 ↑0.0 | 38.7 ↑0.0 | 85.9 ↑0.0 | 37.8 ↑0.0 |
| | VCD | 88.6 ↑1.5 | 46.4 ↑7.2 | 89.2 ↑2.5 | 44.4 ↑5.7 | 87.7 ↑1.8 | 40.6 ↑2.8 |
| | SID | 87.9 ↑0.8 | 42.4 ↑3.2 | 87.2 ↑0.5 | 42.5 ↑3.8 | 86.5 ↑0.6 | 39.9 ↑2.1 |
| | PBA | 87.6 ↑0.5 | 40.2 ↑1.0 | 90.2 ↑3.5 | 45.7 ↑7.0 | 87.3 ↑1.4 | 41.5 ↑3.7 |
| | OLM | 89.6 ↑2.5 | 44.2 ↑5.0 | 90.0 ↑3.3 | 48.8 ↑10.1 | 88.2 ↑2.3 | 43.8 ↑6.0 |
| Popular | Greedy | 85.8 ↑0.0 | 40.4 ↑0.0 | 86.0 ↑0.0 | 39.4 ↑0.0 | 85.6 ↑0.0 | 38.2 ↑0.0 |
| | VCD | 86.2 ↑0.4 | 48.8 ↑8.4 | 87.3 ↑1.3 | 46.3 ↑6.9 | 87.1 ↑1.5 | 41.2 ↑3.0 |
| | SID | 85.1 ↓0.7 | 45.1 ↑4.7 | 85.1 ↓0.9 | 44.6 ↑5.2 | 85.3 ↓0.3 | 39.8 ↑1.6 |
| | PBA | 86.2 ↑0.4 | 41.6 ↑1.2 | 88.4 ↑2.4 | 47.5 ↑8.1 | 86.8 ↑1.2 | 42.3 ↑4.1 |
| | OLM | 87.3 ↑1.5 | 46.5 ↑6.1 | 88.6 ↑2.6 | 50.2 ↑10.8 | 87.4 ↑1.8 | 44.8 ↑6.6 |
| Adversarial | Greedy | 83.6 ↑0.0 | 42.6 ↑0.0 | 84.3 ↑0.0 | 41.0 ↑0.0 | 84.0 ↑0.0 | 39.7 ↑0.0 |
| | VCD | 81.9 ↓1.7 | 53.1 ↑10.5 | 83.8 ↓0.5 | 49.7 ↑8.7 | 84.5 ↑0.5 | 43.7 ↑4.0 |
| | SID | 82.3 ↓1.3 | 47.9 ↑5.3 | 82.9 ↓1.4 | 46.9 ↑5.9 | 83.2 ↓0.8 | 42.5 ↑2.8 |
| | PBA | 83.7 ↑0.1 | 44.0 ↑1.4 | 84.5 ↑0.2 | 51.3 ↑10.3 | 84.1 ↑0.1 | 45.2 ↑5.5 |
| | OLM | 83.6 ↑0.0 | 50.1 ↑7.5 | 83.9 ↓0.4 | 54.9 ↑13.9 | 84.8 ↑0.8 | 48.4 ↑8.7 |

**Results and Analysis on POPE.** The experimental results, presented in Table 3, show that PBA achieves an even greater performance improvement than SID, while OLM surpasses VCD. Prediction accuracy increases as the output distribution approaches balance. However, it is worth noting that on the Adversarial subset, when the distribution shifts beyond balance and biases toward "*Yes*," accuracy begins to decline, aligning with the conclusion in Section 4.3. Although PBA and OLM are not designed for hallucination mitigation, they produce results similar to contrastive decoding methods, suggesting that contrastive decoding does not effectively address hallucinations. For the experimental results on the AOKVQA and GQA datasets, please refer to Section E.

Table 4: Performance of contrastive decoding methods on other benchmarks using greedy search.

| Method | MME-P ↑ | MME-C ↑ | CHAIR-S ↓ | CHAIR-I ↓ | Nocaps ↑ | LLaVA-Bench ↑ |
|---|---|---|---|---|---|---|
| Greedy | 1475.1 ↑00.0 | 349.6 ↑00.0 | 21.6 ↓0.0 | 7.2 ↓0.0 | 83.8 ↑0.0 | 65.7 ↑0.0 |
| VCD | 1389.9 ↓85.2 | 294.6 ↓55.0 | 25.4 ↑3.8 | 8.1 ↑0.9 | 81.2 ↓2.6 | 64.6 ↓1.1 |
| SID | 1396.4 ↓78.7 | 304.1 ↓45.5 | 24.9 ↑3.3 | 7.5 ↑0.3 | 81.9 ↓1.9 | 64.9 ↓0.8 |

**Results and Analysis on Other Benchmarks.** As shown in Table 4, for the discriminative task MME, contrastive decoding methods did not yield any performance gains, as the model's output

did not exhibit an initial distributional bias. In fact, it led to a decline in performance. Similarly, in generative tasks such as CHAIR, Nocaps, and LLaVA-Bench, contrastive decoding offered no improvements. This clearly demonstrates that contrastive decoding can only enhance performance in discriminative tasks where the initial output distribution is skewed towards "*No.*" However, such improvements do not fundamentally address the issue of object hallucination.

## 5.2 Standalone Application of the Constraint.

The second misleading factor contributing to performance improvement is that the adaptive plausibility constraint degrades the sampling strategy into a greedy search strategy.

To investigate this, we plan to apply the adaptive plausibility constraint in isolation while using sampling as the decoding strategy. This will demonstrate the significant performance gains that occur when the constraint forces the sampling strategy to behave like greedy search. When the adaptive plausibility constraint is applied independently, the model's output distribution can be defined as:

$$y_t \sim p_\theta\left(y_t \mid v, x, y_{<t}\right) \propto \exp\left(\mathrm{logit}_\theta\left(y_t \mid v, x, y_{<t}\right)\right), y_t \in \mathcal{V}_{\mathrm{head}}(y_{<t}) \tag{5}$$

**Experimental Settings.** We utilize QwenVL-7B, LLaVA-v1.5-7B, and LLaVA-v1.5-13B as our foundational MLLMs, employing a sampling decoding strategy. Regarding the critical POPE benchmark, our experiments are conducted on the GQA dataset, where the original output distribution of MLLMs is relatively balanced. Consequently, the modification introduced by contrastive decoding methods, which shifts the output distribution towards "*Yes*," does not introduce a positive bias. However, since the adaptive plausibility constraint converts the sampling strategy into a greedy search, the model's performance still improves.

Table 5: Influence of Independent Application of the Adaptive Plausibility Constraint on Model Performance. **Sample**[†] refers to the sampling strategy that applies the constraint independently.

| Category | Method | LLaVA-v1.5-7B | | LLaVA-v1.5-13B | | QwenVL-Chat-7B | |
| --- | --- | --- | --- | --- | --- | --- | --- |
| | | **Accuracy** | **Yes (%)** | **Accuracy** | **Yes (%)** | **Accuracy** | **Yes (%)** |
| Random | sample | 83.8 ↑0.0 | 45.6 ↑0.0 | 84.6 ↑0.0 | 45.9 ↑0.0 | 81.5 ↑0.0 | 41.1 ↑0.0 |
| | VCD | 86.6 ↑2.8 | 52.5 ↑6.9 | 86.7 ↑2.1 | 49.5 ↑3.6 | 83.8 ↑2.3 | 44.0 ↑2.9 |
| | ICD | 85.2 ↑1.4 | 47.0 ↑1.4 | 85.8 ↑1.2 | 44.9 ↓1.0 | 82.5 ↑1.0 | 42.0 ↑0.9 |
| | SID | 84.9 ↑1.1 | 49.1 ↑3.5 | 86.0 ↑1.4 | 49.8 ↑3.9 | 82.9 ↑1.4 | 43.5 ↑2.4 |
| | sample[†] | **85.4 ↑1.6** | **45.1 ↓0.5** | **86.1 ↑1.5** | **45.3 ↓0.6** | **83.0 ↑1.5** | **41.8 ↑0.7** |
| Popular | sample | 77.3 ↑0.0 | 52.1 ↑0.0 | 80.6 ↑0.0 | 49.9 ↑0.0 | 76.8 ↑0.0 | 46.1 ↑0.0 |
| | VCD | 78.7 ↑1.4 | 59.4 ↑7.3 | 82.9 ↑2.3 | 52.4 ↑2.5 | 78.2 ↑1.4 | 49.4 ↑3.3 |
| | ICD | 78.1 ↑0.8 | 54.0 ↑1.9 | 81.5 ↑0.9 | 49.3 ↓0.6 | 77.5 ↑0.7 | 47.2 ↑1.1 |
| | SID | 78.4 ↑1.1 | 53.7 ↑1.6 | 82.5 ↑1.9 | 53.3 ↑3.4 | 77.9 ↑1.1 | 48.0 ↑1.9 |
| | sample[†] | **78.6 ↑1.3** | **52.0 ↓0.1** | **81.8 ↑1.2** | **49.6 ↓0.3** | **78.1 ↑1.3** | **46.8 ↑0.7** |
| Adversarial | sample | 75.1 ↑0.0 | 54.1 ↑0.0 | 78.2 ↑0.0 | 53.2 ↑0.0 | 76.4 ↑0.0 | 45.5 ↑0.0 |
| | VCD | 76.4 ↑1.3 | 62.5 ↑8.4 | 80.3 ↑2.1 | 57.0 ↑3.8 | 78.6 ↑2.2 | 49.2 ↑3.7 |
| | ICD | 75.8 ↑0.7 | 54.2 ↑0.1 | 79.2 ↑1.0 | 52.8 ↓0.4 | 76.8 ↑0.4 | 46.0 ↑0.5 |
| | SID | 76.3 ↑1.2 | 57.5 ↑3.4 | 78.7 ↑0.5 | 57.5 ↑4.3 | 77.2 ↑0.8 | 47.5 ↑2.0 |
| | sample[†] | **76.3 ↑1.2** | **54.2 ↑0.1** | **79.5 ↑1.3** | **53.1 ↓0.1** | **77.9 ↑1.5** | **46.2 ↑0.7** |

**Results and Analysis on POPE.** The experimental results in Table 5 show that when MLLMs adopt sampling as the decoding strategy, applying the adaptive plausibility constraint alone yields an approximate 2.5% performance improvement, effectively validating the conclusion in Section 4.3. Notably, since the adaptive plausibility constraint is entirely unrelated to hallucination mitigation yet achieves performance on par with various contrastive decoding methods, this strongly suggests that contrastive decoding methods do not actually mitigate hallucinations. For the experimental results on the AOKVQA and COCO datasets, please refer to Section F.

**Results and Analysis on Other Benchmarks.** As shown in Table 6, across all other benchmarks, employing the adaptive plausibility constraint alone under direct sampling yields the most significant performance gains. This suggests that the main reason behind the performance improvements observed with contrastive decoding lies in the fact that the constraint reduces direct sampling to greedy search. Crucially, this mechanism is unrelated to the original goal of mitigating hallucination.

Table 6: Performance of contrastive decoding methods on other benchmarks using direct sampling.

| Method | MME-P ↑ | MME-C ↑ | CHAIR-S ↓ | CHAIR-I ↓ | Nocaps ↑ | LLaVA-Bench ↑ |
|--------|---------|---------|-----------|-----------|----------|---------------|
| Sample | 1282.1 ↑0.0 | 286.7 ↑0.0 | 23.2 ↓0.0 | 7.7 ↓0.0 | 81.4 ↑0.0 | 64.3 ↑0.0 |
| Sample† | **1392.4** ↑110.3 | **302.6** ↑15.9 | **22.4** ↓0.8 | **7.4** ↓0.3 | **83.1** ↑1.7 | **65.3** ↑1.0 |
| VCD | 1361.7 ↑79.6 | 278.6 ↓8.1 | 25.8 ↑2.6 | 8.3 ↑0.6 | 80.9 ↓0.5 | 64.2 ↓0.1 |
| SID | 1364.6 ↑82.5 | 284.1 ↓2.6 | 25.2 ↑2.0 | 7.9 ↑0.2 | 81.2 ↑0.2 | 64.5 ↑0.2 |

## 6 Discussion on Hallucination Mitigation

Based on the insights from Sections 4 and 5, we propose some new criteria for evaluating object hallucination mitigation in MLLMs.

**Impact of Decoding Strategies**. When evaluating on POPE, it is essential to account for the substantial influence of different decoding strategies on model performance. Notably, greedy search consistently outperforms sampling-based methods such as direct sampling and nucleus sampling. If a hallucination mitigation method involves modifications to the sampling module, careful consideration must be given to whether these changes affect the core properties of the decoding strategy.

**Avoiding Unidirectional Modification**. When evaluating hallucination mitigation methods, it is essential to assess whether they alter responses unidirectionally. Given the skewed output distribution of MLLMs across multiple datasets, a method that merely rebalances responses may create the illusion of improved performance. However, such adjustments do not genuinely mitigate hallucinations.

**Balancing Correction and Preservation**. An effective hallucination mitigation method must strike a balance: it should correct incorrect answers while preserving correct ones. However, as shown in Figure 2, some flawed approaches, despite fixing many errors, also introduce unnecessary modifications to originally correct responses. This behavior resembles mere answer editing rather than genuine hallucination mitigation. To enhance evaluation rigor, future studies should explicitly report instances where correct responses are mistakenly altered, providing a clearer measure of a method's true effectiveness.

## 7 Conclusion

This study demonstrates that the performance improvements of contrastive decoding on the POPE benchmark largely stem from two misleading factors: (1) a unidirectional shift in the model's output distribution, which biases it toward generating "*Yes*" responses, artificially balancing the distribution in certain datasets, and (2) the adaptive plausibility constraint, which reduces sampling decoding to greedy search. By comparing experimental results from spurious improvement methods and contrastive decoding, we confirm that while contrastive decoding enhances performance, it ultimately fails to mitigate hallucinations.

## Impact Statement

The broader impact of this work includes fostering more transparent and accountable AI systems, particularly in applications where misinformation can have serious ethical and societal consequences, such as healthcare, legal reasoning, and scientific discovery. Our analysis underscores the importance of critical evaluation in AI research to prevent the deployment of methods that may not work as intended. While our work does not introduce new risks, it serves as a cautionary study that helps guide future research toward more robust and responsible AI development.

## Acknowledgements

This work is supported by the National Natural Science Foundation of China under Grant 62176246. This work is also supported by Anhui Province Key Research and Development Plan (202304a05020045), Anhui Province Natural Science Foundation (2208085UD17) and National Natural Science Foundation of China under Grant 62406098.

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

# A Contrastive Decoding for Hallucinations

This section details the components and workflows of three mainstream hallucination mitigation methods: VCD, ICD, and SID. These techniques employ contrastive decoding strategies to reduce hallucinatory content, ensuring outputs align more closely with the visual input.

## A.1 Visual Contrastive Decoding

Visual Contrastive Decoding (VCD) acts as a corrective mechanism, reducing hallucinations by contrasting with distributions derived from distorted visual inputs. Specifically, for a given textual query $x$ and a visual input $v$, the model generates two distinct output distributions: one conditioned on the original $v$, and the other on a distorted version $v'$. The distorted input $v'$ is derived by applying predefined perturbations (e.g., a Gaussian noise mask) to $v$. Subsequently, a new contrastive probability distribution is computed by leveraging the differences between these two distributions. This contrastive distribution, denoted as $p_{vcd}$, is defined as:

$$p_{vcd}\left(y \mid v, v', x\right) = \text{softmax}\left[\text{logit}_\theta\left(y \mid v, x\right) + \alpha \cdot \left(\text{logit}_\theta\left(y \mid v, x\right) - \text{logit}_\theta\left(y \mid v', x\right)\right)\right], \quad (6)$$

A challenge in this process is avoiding indiscriminate penalization of the entire output space, as this could unfairly suppress valid predictions while encouraging the generation of implausible tokens. To address this, VCD integrates an **adaptive plausibility constraint**. This constraint dynamically adjusts penalization based on the confidence levels inferred from the output distribution conditioned on the original visual input $v$. The constraint is defined as follows:

$$\mathcal{V}_{\text{head}}\left(y_{<t}\right) = \left\{y_t \in \mathcal{V} \mid p_\theta\left(y_t \mid v, x, y_{<t}\right) \geq \beta \max_w p_\theta\left(w \mid v, x, y_{<t}\right)\right\},$$

$$p_{vcd}\left(y_t \mid v, v', x\right) = 0 \quad \text{if } y_t \notin \mathcal{V}_{\text{head}}\left(y_{<t}\right)$$

$$(7)$$

where $\mathcal{V}$ denotes the output vocabulary of MLLMs, and $\beta$ is a hyperparameter that controls the truncation of the next-token distribution. Larger values of $\beta$ enforce more aggressive truncation, retaining only the tokens with the highest probabilities.

By integrating the contrastive adjustment with the adaptive plausibility constraint, the complete formulation is expressed as follows:

$$y_t \sim \text{softmax}\left[(1 + \alpha) \cdot \text{logit}_\theta\left(y_t \mid v, x, y_{<t}\right) - \alpha \cdot \text{logit}_\theta\left(y_t \mid v', x, y_{<t}\right)\right],$$

$$\text{subject to} \quad y_t \in \mathcal{V}_{\text{head}}(y_{<t}).$$

$$(8)$$

## A.2 Instruction Contrastive Decoding

Based on findings that instruction disturbances with negative prefixes significantly amplify hallucinations by increasing multimodal alignment uncertainty, Instruction Contrastive Decoding (ICD) mitigates hallucinations by initially emphasizing the probabilities of hallucinated concepts and subsequently detaching these from the original probability distribution. Accordingly, the contrastive distribution, $p_{icd}$, can be defined as:

$$p_{icd}\left(y \mid v, x, x'\right) = \text{softmax}\left[\text{logit}_\theta\left(y \mid v, x\right) - \lambda \cdot \text{logit}_\theta\left(y \mid v, x'\right)\right]. \quad (9)$$

A larger $\lambda$ imposes a stronger penalty on the decisions made by MLLMs under disturbances. Here, $x'$ represents perturbed instructions involving negative prefixes. Additionally, ICD integrates **adaptive plausibility constraint** from VCD to prevent the unjust suppression of valid predictions.

## A.3 Self-Introspective Decoding

Building on VCD and ICD, SID recognizes that directly perturbing the entire original input introduces excessive uncertainty and noise, hindering the induction of the desired hallucination effect. To address this, as shown on the far right of Figure 1, SID adjusts the model architecture by retaining only a small subset of image tokens with low attention scores after the early decoder layers. This

adaptive mechanism enhances the generation of vision-and-text association hallucinations during auto-regressive decoding. Subsequently, SID isolates these hallucinations from the original probability distribution, leading to the definition of the contrastive distribution $p_{sid}$ as:

$$p_{sid}\left(y \mid v, x\right) = \operatorname{softmax}\left[\operatorname{logit}_{\theta}\left(y \mid v, x\right) + \alpha \cdot \left(\operatorname{logit}_{\theta}\left(y \mid v, x\right) - \operatorname{logit}_{\theta'}\left(y \mid v, x\right)\right)\right]. \quad (10)$$

Here, $\theta'$ represents the MLLM with structural modifications introduced by SID. Additionally, SID incorporates the **adaptive plausibility constraint**.

## B  Additional Research on Changes in Prediction

After applying Visual Contrastive Decoding, the prediction shifts of the LLaVA-v1.5-7B model are shown in Figure 6. It is evident that across all datasets, the number of samples transitioning from Positive to Negative is significantly smaller than those shifting from Negative to Positive. This indicates that the model's output distribution is biased towards *Yes*.

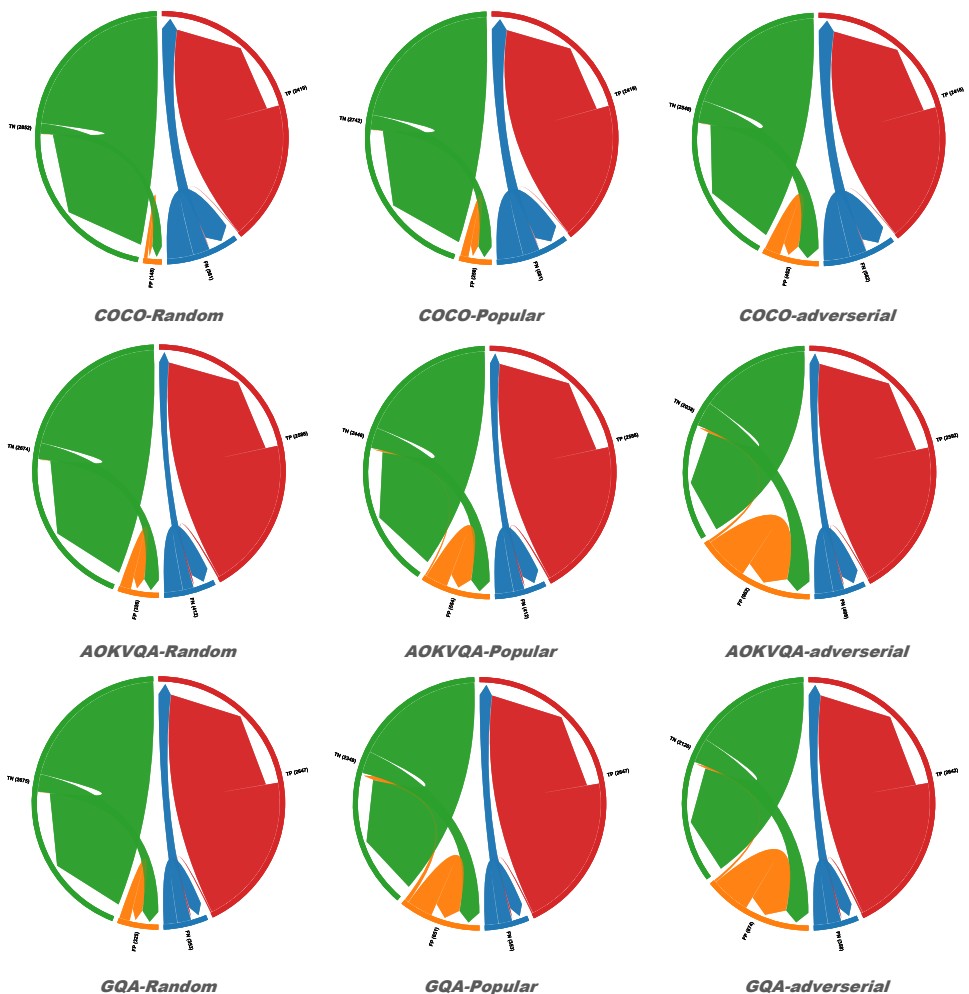

Figure 6: Changes in the distribution of model predictions across all datasets after applying Visual Contrastive Decoding.

## C  The True Impact of Contrastive Perturbations on Prediction Outcomes

We begin by presenting our conclusion: image perturbations in *visual contrastive decoding* can cause the model to randomly "overlook" certain elements of the image due to their stochastic nature.

Consequently, the model's response is based on only a partial view of the input. This phenomenon is illustrated with a concrete example, as shown in Figure 7.

Question: How many uncut fruits are in the image?

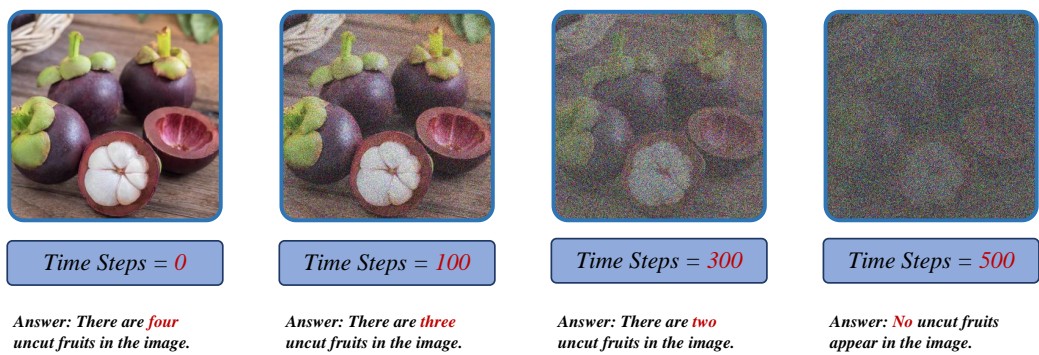

Figure 7: Visualization of the Actual Impact of Contrastive Perturbations.

With the original visual input, the model exhibited a hallucination, responding: "There are four uncut fruits in the image." After introducing diffusion noise and setting the diffusion time steps to 100, 300, and 500 respectively, the LLaVA-v1.5-7B model produced the following responses: "There are 3 / 2 / 0 uncut fruits in the image," correspondingly. These results demonstrate that the presence of noise causes the model to lose access to certain localized visual details, leading it to respond based on the remaining visible features.

This also helps explain why, in POPE benchmark, contrastive examples often cause the model to answer "*No*", not due to external bias, but because it genuinely fails to detect the relevant object. However, this effect should not be mistaken for hallucination. In fact, contrastive samples do not induce hallucinations. For example, when *visual contrastive decoding* is applied to this case, the model correctly outputs: "There are four uncut fruits in the image."

This brings us back to the foundational assumption behind contrastive decoding methods for mitigating hallucinations. These approaches are grounded in the belief that object hallucinations in MLLMs primarily stem from overly dominant linguistic priors, and thus attempt to suppress such priors through contrastive mechanisms. However, in practice, hallucinations often arise because MLLMs fail to correctly ground specific semantic concepts in the visual input, for instance, the concept of "uncut" in the previous example. As such, the failure of *visual contrastive decoding* can be attributed to a fundamental flaw in its underlying assumption.

## D  Performance across different decoding strategies.

Table 7 presents the impact of various decoding strategies on the prediction performance of LLaVA-v1.5-7B evaluated on the POPE Benchmark. It is evident that the greedy strategy yields significantly better results compared to the sampling strategy.

Table 7: Performance on the MSCOCO dataset across different decoding strategies.

| Model | Decoding | Random | Popular | Adversarial |
|---|---|---|---|---|
| LLaVA | Greedy | 87.1 ↓ 0.0 | 85.8 ↓ 0.0 | 83.6 ↓ 0.0 |
| | Sample | 83.6 ↓ 3.5 | 82.4 ↓ 3.4 | 80.2 ↓ 3.4 |
| QwenVL | Greedy | 86.0 ↓ 0.0 | 85.6 ↓ 0.0 | 84.0 ↓ 0.0 |
| | Sample | 85.2 ↓ 0.8 | 84.2 ↓ 1.4 | 82.3 ↓ 1.7 |

# E Further Experiments on Forced Distribution Adjustment

Tables 8 and 9 present the performance of Prompt-Based Adjustment (PBA) and Output Layer Modification (OLM) on the AOKVQA and GQA datasets. On both datasets, PBA and OLM maintain performance levels comparable to contrastive decoding methods. However, after surpassing the balanced distribution, accuracy declines rather than improving.

Table 8: Performance of Prompt-Based Adjustment (PBA) and Output Layer Modification (OLM) on AOKVQA dataset

| Category | Decoding | LLaVA-v1.5-7B | | | LLaVA-v1.5-13B | | |
|---|---|---|---|---|---|---|---|
| | | Accuracy | F1-Score | Yes(%) | Accuracy | F1-Score | Yes(%) |
| Random | Greedy | 88.6 | 88.1 | 45.1 | 88.8 | 88.1 | 44.1 |
| | VCD | 86.8 | 87.0 | 52.0 | 87.7 | 87.4 | 47.8 |
| | SID | 88.6 | 88.5 | 48.7 | 87.6 | 87.4 | 48.5 |
| | PBA | 89.0 | 88.5 | 45.9 | 88.7 | 89.0 | 52.8 |
| | OLM | 89.0 | 89.1 | 51.3 | 87.3 | 87.9 | 55.9 |
| Popular | Greedy | 85.2 | 85.0 | 48.5 | 86.7 | 86.2 | 46.2 |
| | VCD | **82.6** | 83.6 | **56.2** | **85.5** | 85.5 | 50.0 |
| | SID | **84.1** | 84.6 | 53.2 | **85.1** | 85.3 | 51.0 |
| | PBA | **84.8** | 84.8 | 50.1 | **83.6** | 84.8 | 57.9 |
| | OLM | **82.6** | 83.8 | **57.7** | **83.6** | 85.1 | **59.5** |
| Adversarial | Greedy | 78.8 | 79.8 | 54.9 | 80.3 | 80.8 | 52.6 |
| | VCD | **75.5** | 78.4 | **63.6** | **79.4** | 80.7 | **56.4** |
| | SID | **77.8** | 79.7 | 59.1 | **79.2** | 80.6 | 57.4 |
| | PBA | **78.3** | 79.6 | 56.6 | **74.7** | 78.3 | 66.8 |
| | OLM | **75.0** | 78.3 | **65.3** | **74.8** | 78.7 | **68.3** |

Table 9: Performance of Prompt-Based Adjustment (PBA) and Output Layer Modification (OLM) on GQA dataset

| Category | Decoding | LLaVA-v1.5-7B | | | LLaVA-v1.5-13B | | |
|---|---|---|---|---|---|---|---|
| | | Accuracy | F1-Score | Yes(%) | Accuracy | F1-Score | Yes(%) |
| Random | Greedy | 89.4 | 88.9 | 45.5 | 89.5 | 88.9 | 45.1 |
| | VCD | 88.0 | 88.4 | 53.6 | 88.1 | 88.0 | 49.7 |
| | SID | 88.8 | 88.7 | 49.1 | 88.9 | 88.8 | 49.1 |
| | PBA | 89.4 | 89.0 | 47.0 | 88.8 | 89.2 | 53.9 |
| | OLM | 89.4 | 89.7 | 52.9 | 87.1 | 87.9 | 56.9 |
| Popular | Greedy | 84.0 | 84.2 | 50.8 | 86.4 | 86.1 | 48.1 |
| | VCD | **82.5** | 83.9 | **59.1** | **85.7** | 85.5 | **52.1** |
| | SID | **82.9** | 83.8 | **55.0** | **84.8** | 85.2 | **53.2** |
| | PBA | **83.2** | 83.7 | **53.1** | **84.8** | 84.8 | **61.9** |
| | OLM | **79.8** | 82.0 | **62.5** | **80.8** | 83.1 | **63.1** |
| Adversarial | Greedy | 80.9 | 81.7 | 53.9 | 82.2 | 82.6 | 52.3 |
| | VCD | **78.0** | 80.6 | 63.3 | **81.2** | 82.4 | **56.8** |
| | SID | **79.9** | 81.3 | 57.8 | **81.1** | 82.4 | **57.6** |
| | PBA | 80.5 | 81.5 | 55.9 | **81.1** | 82.3 | **67.3** |
| | OLM | **76.4** | 79.6 | 65.9 | **75.9** | 79.6 | **68.3** |

# F Further Experiments on Standalone Application of the Constraint

Tables 10 and 11 present the performance of the adaptive plausibility constraint on the AOKVQA and COCO datasets. When applied independently, the adaptive plausibility constraint consistently improves performance by 1.5% to 2% across both datasets.

Table 10: Impact of Adaptive Plausibility Constraint (Applied Independently) on AOKVQA Dataset

| Category | Decoding | LLaVA-v1.5-7B | | | LLaVA-v1.5-13B | | |
| --- | --- | --- | --- | --- | --- | --- | --- |
| | | Accuracy | F1-Score | Yes(%) | Accuracy | F1-Score | Yes(%) |
| Random | sample | 84.6 | 83.9 | 45.4 | 84.7 | 83.9 | 45.3 |
| | VCD | 85.9 | 86.1 | 51.7 | 86.7 | 86.5 | 48.4 |
| | ICD | 86.5 | 85.8 | 45.3 | 86.3 | 85.5 | 44.5 |
| | SID | 86.8 | 86.6 | 48.8 | 85.8 | 85.5 | 48.3 |
| | sample$^\dagger$ | **86.5** | **85.8** | **45.3** | **86.6** | **85.8** | **44.8** |
| Popular | sample | 80.3 | 80.2 | 49.7 | 81.8 | 81.5 | 48.2 |
| | VCD | 81.3 | 82.4 | 56.2 | 84.1 | 84.2 | 51.0 |
| | ICD | 82.2 | 82.1 | 49.6 | 83.4 | 82.9 | 47.4 |
| | SID | 82.9 | 83.3 | 52.7 | 82.9 | 83.1 | 51.2 |
| | sample$^\dagger$ | **82.2** | **82.1** | **49.6** | **83.6** | **83.2** | **47.8** |
| Adversarial | sample | 74.8 | 76.2 | 55.9 | 77.0 | 77.9 | 54.0 |
| | VCD | 74.8 | 77.6 | 62.8 | 78.4 | 79.7 | 56.4 |
| | ICD | 76.1 | 77.5 | 56.4 | 77.5 | 78.3 | 53.5 |
| | SID | 76.8 | 78.6 | 58.4 | 77.9 | 79.4 | 57.4 |
| | sample$^\dagger$ | **76.1** | **77.5** | **56.4** | **77.9** | **78.7** | **54.0** |

Table 11: Impact of Adaptive Plausibility Constraint (Applied Independently) on COCO Dataset

| Category | Decoding | LLaVA-v1.5-7B | | | LLaVA-v1.5-13B | | |
| --- | --- | --- | --- | --- | --- | --- | --- |
| | | Accuracy | F1-Score | Yes(%) | Accuracy | F1-Score | Yes(%) |
| Random | sample | 83.6 | 81.8 | 39.8 | 83.9 | 82.3 | 40.8 |
| | VCD | 87.8 | 87.3 | 46.4 | 87.8 | 87.1 | 44.6 |
| | ICD | 85.4 | 83.7 | 39.8 | 85.5 | 83.9 | 40.3 |
| | SID | 86.6 | 85.4 | 41.9 | 86.4 | 85.3 | 42.7 |
| | sample$^\dagger$ | **85.4** | **83.7** | **39.8** | **85.5** | **84.0** | **40.6** |
| Popular | sample | 82.4 | 80.7 | 41.0 | 83.0 | 81.5 | 41.7 |
| | VCD | 85.8 | 85.5 | 48.4 | 86.2 | 85.7 | 46.1 |
| | ICD | 84.1 | 82.5 | 41.1 | 84.6 | 83.1 | 41.2 |
| | SID | 83.6 | 82.7 | 44.9 | 84.4 | 83.5 | 44.8 |
| | sample$^\dagger$ | **84.1** | **82.5** | **41.1** | **84.6** | **83.1** | **41.6** |
| Adversarial | sample | 80.2 | 78.7 | 43.2 | 81.2 | 79.9 | 43.5 |
| | VCD | 81.1 | 81.7 | 52.9 | 83.0 | 82.9 | 49.3 |
| | ICD | 81.8 | 80.5 | 43.3 | 82.6 | 81.3 | 43.2 |
| | SID | 80.9 | 80.4 | 47.5 | 81.8 | 81.3 | 47.3 |
| | sample$^\dagger$ | **81.8** | **80.5** | **43.3** | **82.7** | **81.5** | **43.4** |

