# OpenReview forum: "The Mirage of Performance Gains: Why Contrastive Decoding Fails to Mitigate Object Hallucinations in MLLMs?"
_NeurIPS.cc/2025/Conference — NeurIPS 2025 poster_

### Official Review · Reviewer_bPyU · 2025-06-25

**Clarity:** 3
**Significance:** 3
**Originality:** 3
**Rating:** 5
**Confidence:** 3

**Summary:**

This paper critically challenges the perceived effectiveness of contrastive decoding strategies for mitigating object hallucinations in MLLMs. This paper claims that the gained performance on the POPE benchmark are highly misleading and do not represent genuine hallucination mitigation. It attribute these "spurious improvements" to two primary factors: (1) a unidirectional shift of the model's output distribution towards "yes" response, and (2) the adaptive plausibility constraint, which effectively degrades sampling decoding to greedy search. To substantiate their claims, the paper introduces and evaluates "spurious improvement methods" that achieve comparable performance to contrastive decoding, decisively demonstrating that the reported gains are unrelated to actual hallucination mitigation.

**Questions:**

Please refer to the weakness below.

**Ethical Concerns:**

["NO or VERY MINOR ethics concerns only"]

**Final Justification:**

My comments are properly addressed. The authors show significant efforts on the newly added experimentations. As a result, I increase my rating. But I hope that the modifications, such as details of more complex hallucinations, added in rebuttal can be included in the revised version, which are important for the completeness of limitation discussion.

**Limitations:**

As the paper does not introduce a new method, there is no discussion of methodological limitations.

**Quality:**

3

**Strengths And Weaknesses:**

### Strengths
1. The paper critically challenges the contrastive decoding paradigm in MLLM hallucination mitigation, providing evidence that the observed performance gains are deceptive. This critique helps explore the actual causes of alleviating hallucinations for the community.
2. The authors include extensive analyses covering ablation studies on both hallucination and general benchmarks between tricky designs and original contrastive decoding from the perspective of unidirectional shift and adaptive plausibility constraint.

### Weaknesses
1. The paper criticizes contrastive decoding, but it mostly focuses on older methods. It could be better to analyze on newer methods [1]. [1] Park, Yeji, et al. "Convis: Contrastive decoding with hallucination visualization for mitigating hallucinations in multimodal large language models." Proceedings of the AAAI Conference on Artificial Intelligence. Vol. 39. No. 6. 2025.
2. The paper's definition and verification of object hallucination are insufficient, and it does not provide explanations or experimental results for whether more complex relational or attribute hallucinations are similarly unresolved by contrastive decoding.
3. Despite using LLaVA-v1.5 and QwenVL-7B, the experimental scope for this training-free method could be expanded to include more recent, larger, and architecturally diverse MLLMs (e.g., Qwen-2.5-VL, LLaVA-Next) to ensure broader applicability of its conclusions.

---

> ### Author Rebuttal · Authors · 2025-07-31
>
> # *Initial Acknowledgment*
>
> **We are grateful to the reviewers for their valuable and encouraging feedback. The insightful suggestions and comments helped us identify areas for clarification and further improvement. In the following, we address each of the reviewers’ concerns point by point.**
>
> ---
>
> # *W1: The paper criticizes contrastive decoding, but it mostly focuses on older methods. It could be better to analyze on newer methods [1].*
>
> **In fact, Self-Introspective Decoding (SID) presented in our paper was published at ICLR 2025 and, like Convis, is a contemporaneous work and among the most recent approaches.** However, we understand your interest in more recent comparative decoding methods. **Therefore, we present experimental results of Convis [1] here.**
>
> **We report the performance of Convis on the MSCOCO dataset from the POPE benchmark in the tables below.** Convis shows an output distribution similar to VCD, heavily biased toward YES. This bias enhances performance on the random and popular subsets but reduces it on the adversarial subset. Despite this, our two dummy methods, PBA and OLM, achieve prediction accuracy gains comparable to those of Convis.
>
> ***Table: Changes in Output Distribution (Yes%) Induced by Convis***
> | **Method**      | **Random**   | **Popular**   | **Adversarial** |
> |:--------------:|:------------:|:-------------:|:----------------:|
> | Greedy   | 39.2 ↑0.0    | 40.4 ↑0.0     | 42.6 ↑0.0        |
> | VCD   | 46.4 ↑7.2    | 48.8 ↑8.4     | 53.1 ↑10.5       |
> | Convis   | 43.4 ↑4.2    | 45.7 ↑5.3     | 48.8 ↑6.2        |
> | PBA    | 40.2 ↑1.0    | 41.6 ↑1.2     | 44.0 ↑1.4        |
> | OLM   | 44.2 ↑5.0    | 46.5 ↑6.1     | 50.1 ↑7.5        |
>
> ***Table: Prediction Accuracy of Convis***
> | **Method** | **Random**| **Popular**| **Adversarial** |
> |:--------:|:--------:|:---------:|:--------:|
> | Greedy  | 87.1 ↑0.0    | 85.8 ↑0.0     | 83.6 ↑0.0   |
> | VCD   | 88.6 ↑1.5    | 86.2 ↑0.4     | 81.9 ↓1.7   |
> | Convis   | 88.1 ↑1.0    | 85.9 ↑0.1     | 82.1 ↓1.5  |
> | PBA    | 87.6 ↑0.5    | 86.2 ↑0.4     | 83.7 ↑0.1  |
> | OLM   | 89.6 ↑2.5    | 87.3 ↑1.5     | 83.6 ↑0.0  |
>
> **The following table presents the performance of Convis on other datasets, including MME, CHAIR, NoCaps, and LLaVA-Bench.**  It can be observed that, similar to methods like VCD, ICD, and SID, the prediction accuracy decreases after applying Convis.
>
> | **Method** | **MME-P↑** | **MME-C↑** | **CHAIR-S↓** | **CHAIR-I↓** | **Nocaps↑** | **LLaVA-Bench↑** |
> |:----------:|:---------:|:---------:|:-----------:|:-----------:|:----------:|:----------------:|
> | Greedy | **1475.1**    | **349.6**     | **21.6**    | **7.2**     | **83.8**     | **65.7**       |
> | Convis    | 1343.6    | 306.3     | 24.1        | 8.7     | 81.7       | 64.9     |
>
>
> The experimental results above demonstrate that, like VCD, ICD, and SID, Convis does not effectively mitigate hallucination.
>
> ---
>
> # *W2: The paper's definition and verification of object hallucination are insufficient, and it does not provide explanations or experimental results for whether more complex relational or attribute hallucinations are similarly unresolved by contrastive decoding.*
>
> ## *1. Definition of Object Hallucination*
>
> We appreciate your attention to the definition of object hallucination. **However, our definition is consistent with that adopted in the original papers of methods such as VCD.** Specifically, the definition of object hallucination in the VCD paper is as follows:
>
> > Object hallucination, in this context, refers to the phenomenon where LVLMs generate textual content that is semantically coherent but inconsistent with the ground-truth objects present in the given image.
>
> Since methods such as VCD claim to address this type of object hallucination, we aim to provide a point-by-point response in our paper to demonstrate that they, in fact, fail to resolve such hallucinations.
>
> ## *2. Whether relational or attribute hallucinations also remain unresolved?*
>
> **In fact, our experiments on the MME and LLaVA-Bench datasets, as reported in Tables 4 and 6 of the paper, include evaluations of both relational and attribute hallucinations.** Due to space constraints, we only presented the overall scores for these datasets and omitted the detailed results for individual subsets. However, several of these subsets are specifically designed to assess relational and attribute hallucinations. We present the corresponding subset-level results below.
>
> | **Method**  | **MME-Existence** | **MME-Count** | **MME-Position** | **MME-Color** | **LLaVA Bench-Complex** |
> |:--------:|:-------:|:-------:|:-------:|:------:|:------:|
> | Greedy  | **190.0**  | **145.0**  | **113.3**   | **170.0**   | **81.2**  |
> | VCD   | 180.0  | 135.0   | 96.7    | 150.0   | 79.4   |
>
> As shown, on subsets such as Color and Position from the MME dataset and the Complex subset from LLaVA-Bench, methods like VCD lead to a noticeable drop in prediction accuracy. This clearly indicates that such methods fail to mitigate relational and attribute hallucinations.
>
> ---
>
> # *W3: Despite using LLaVA-v1.5 and QwenVL-7B, the experimental scope for this training-free method could be expanded to include more recent, larger, and architecturally diverse MLLMs.*
>
> We appreciate your interest in exploring more recent MLLMs. **In accordance with your suggestion, we have conducted additional experiments using the LLaVA-NeXT model.** The tables below present VCD's performance on the COCO dataset within the POPE Benchmark, based on the LLaVA-NeXT-13B model. As observed, the results closely mirror those obtained with LLaVA-v1.5-7B. VCD continues to impose a bias in the model’s output distribution toward answering YES, resulting in improved accuracy on the Random and Popular subsets, but reduced accuracy on the Adversarial subset. Notably, the two dummy methods introduced in the paper, PBA and OLM, continue to yield comparable performance gains with ease.
>
> ***Table: Impact of VCD on Output Distribution (Yes%) in LLaVA-NeXT-13B***
> | **Method** | **Random**   | **Popular**  | **Adversarial** |
> |:------------:|:------------:|:------------:|:-------------:|
> | Greedy | 40.0 ↑0.0    | 41.6 ↑0.0    | 44.3 ↑0.0      |
> | VCD | 47.4 ↑7.4    | 50.0 ↑8.4    | 54.6 ↑10.3     |
> | PBA | 41.8 ↑1.8    | 42.5 ↑0.9    | 44.6 ↑0.3      |
> | OLM | 45.6 ↑5.6    | 47.3 ↑5.7    | 51.1 ↑6.8      |
>
> ***Table: Prediction Accuracy of VCD with LLaVA-NeXT-13B***
> | **Method** | **Random**   | **Popular**  | **Adversarial** |
> |:-----------:|:------------:|:------------:|:-------------:|
> | Greedy | 88.4 ↑0.0      | 87.3 ↑0.0   | 84.8 ↑0.0     |
> | VCD    | 89.6 ↑1.2    | 87.5 ↑0.2    | 82.9 ↓1.9     |
> | PBA   | 89.1 ↑0.7    | 88.0 ↑0.7    | 84.7 ↓0.1    |
> | OLM   | 90.2 ↑1.8    | 87.8 ↑0.5    | 84.1 ↓0.7     |
>
> Below are the experimental results of VCD based on the LLaVA-NeXT-13B model across several datasets, including MME, NoCaps, and LLaVA-Bench. Consistent with the findings on LLaVA-v1.5-7B, the application of the VCD method leads to a noticeable drop in accuracy.
>
> | **Method** | **MME-P↑** | **MME-C↑** | **CHAIR-S↓** | **CHAIR-I↓** | **Nocaps↑** | **LLaVA-Bench↑** |
> |:----------:|:---------:|:---------:|:-----------:|:-----------:|:----------:|:---------------:|
> | Greedy | **1575.0** | **332.0** | **18.6**    | **6.8**     | **97.4**   | **87.1**         |
> | VCD        | 1488.3     | 302.1     | 22.7        | 7.9         | 93.8       | 85.2             |
>
> **Finally, we would like to clarify why our experiments are limited to LLaVA-v1.5 and QwenVL-v1.5.** The original papers proposing methods such as VCD primarily conducted their experiments using these two models. Since the aim of our study is to demonstrate the ineffectiveness of these methods, **we intentionally adopted the exact same experimental settings to ensure a fair and direct comparison.** This approach strengthens the validity and persuasiveness of our conclusions. For this reason, we did not extend our experiments to newer models.
>
> ---
>
> # *Final Acknowledgment*
>
> **We sincerely thank the reviewers once again for their time, effort, and valuable feedback. We hope our responses have resolved the concerns raised, and we are happy to provide additional clarification if needed. We truly appreciate your thoughtful review, and we wish you all the best in your future research endeavors.**
>
> ---
>
> [1] Park, Yeji, et al. "Convis: Contrastive decoding with hallucination visualization for mitigating hallucinations in multimodal large language models." Proceedings of the AAAI Conference on Artificial Intelligence. Vol. 39. No. 6. 2025.

---

> > ### Comment · Reviewer_bPyU · 2025-08-05
> >
> > Thanks for the rebuttal and extensive added experimentations. I will raise the score. But I hope that the modifications added in rebuttal can be included in the revised version for clarity

---

> > > ### Author Response · Authors · 2025-08-05
> > >
> > > Thank you very much for your positive feedback and for considering raising the score. We truly appreciate your thoughtful comments and are glad that the additional experiments and clarifications were helpful.
> > >
> > > We will make sure to incorporate all the modifications introduced during the rebuttal phase into the final revised version to ensure clarity and completeness.
> > >
> > > Thank you again for your valuable time and constructive insights. We wish you all the best in your future research endeavors.

---

### Official Review · Reviewer_RnVz · 2025-06-30

**Clarity:** 3
**Significance:** 3
**Originality:** 3
**Rating:** 4
**Confidence:** 4

**Summary:**

This paper critically examines popular methods for reducing object hallucinations in multimodal language models, including common approaches like Visual Contrastive Decoding (VCD), Instruction Contrastive Decoding (ICD), and Self-Introspective Decoding (SID). The authors argue that although these methods appear to boost performance on the POPE benchmark, these improvements are misleading. The real cause of the better scores is not actual reduction in hallucinations, but rather two side effects: (1) pushing the model’s answers toward “Yes” to balance the data, and (2) using a constraint that effectively turns sampling into greedy search, which helps on simple yes/no tasks. The authors demonstrate this by introducing their own artificial tricks (like always encouraging “Yes” or tweaking the output layer) that achieve similar “improvements” without addressing hallucination at all. The key takeaway is that current contrastive decoding techniques don’t really solve the problem, even if the numbers look better.

**Questions:**

In lines 170–174, the observation that MLLMs tend to generate “NO” is discussed. Is this tendency specific to the POPE benchmark, or does it generalize to other datasets and evaluation protocols? If this is not a general phenomenon, how do the conclusions regarding contrastive decoding’s tendency to produce more “YES” answers remain valid?

**Ethical Concerns:**

["NO or VERY MINOR ethics concerns only"]

**Final Justification:**

I read the rebuttal from the authors and the discussion among other reviewers before the final justification.

**Limitations:**

yes

**Paper Formatting Concerns:**

None.

**Quality:**

3

**Strengths And Weaknesses:**

## Strengths:

- The paper tackles a real and important issue: how to fairly evaluate progress on hallucination in multimodal models, which has real-world safety stakes.

- The experimental setup is well thought out and includes multiple models, benchmarks, and clever “control” tricks to highlight the central flaw, underscoring a potentially vital claim for the community: current contrastive decoding techniques don’t really solve the problem, even if the numbers look better.

- The analysis of why these methods get better numbers (by unbalancing answers or changing the search strategy) is clear and persuasive.


## Weaknesses:

- The paper suffers from some notation and presentation problems that hinder readability. For example, a symbol like p_cd is introduced around Equation (2) without clear definition (presumably referring to a contrastive decoding probability distribution), which may confuse readers. Similarly, some results are hard to interpret due to labeling issues, e.g., Table 6 contains a “Sample*” notation that is never explained, and the annotations in Figure 3 are unclear, making it difficult to follow the figure’s message. These issues suggest the exposition could be more polished and self-contained.

- The main conclusion, “we confirm that while contrastive decoding enhances performance, it ultimately fails to mitigate hallucinations”, remains ambiguous. Could the authors clarify the intended implication and its broader significance?

---

> ### Author Rebuttal · Authors · 2025-07-30
>
> # *Initial Acknowledgment*
>
> **We are grateful to the reviewers for their valuable and encouraging feedback. The insightful suggestions and comments helped us identify areas for clarification and further improvement. In the following, we address each of the reviewers’ concerns point by point.**
>
> ---
>
> # *W1: The paper suffers from some notation and presentation problems that hinder readability.*
>
> Thank you for pointing out these issues. We sincerely apologize for the lack of clarity in some of the markup definitions and expressions, which may have affected readability. Below, we clarify each of the ambiguous parts one by one.
>
> ## *1. A symbol like $p_{cd}$ is introduced around Equation (2) without clear definition, which may confuse readers.*
>
> We apologize for the lack of clarity in our explanation. Your understanding of $p_{cd}$ is absolutely correct, it indeed refers to the probability distribution generated by the contrastive decoding strategy. We will promptly add a clear definition of $p_{cd}$ in the manuscript as soon as we are able to modify the PDF.
>
> ## *2. Table 6 contains a "Sample\*" notation that is never explained.*
>
> Thank you for pointing out this issue. We sincerely apologize for our use of an incorrect superscript symbol in this instance. The correct notation should be *Sample†*, consistent with the definition provided in Table 5, which refers to the sampling strategy that applies the adaptive plausibility constraint independently. We deeply regret any confusion this oversight may have caused.
>
> ## *3. The annotations in Figure 3 are unclear, making it difficult to follow the figure’s message.*
>
> **Here, we provide a more detailed explanation of the information presented in the four subfigures of Figure 3.**
>
> - The top-left subfigure uses accuracy as the unit on the vertical axis, illustrating that using greedy decoding as the strategy yields significantly higher prediction accuracy compared to direct sampling.
>
> - The top-right subfigure presents a specific test example, featuring the question “Is there a cellphone in the image?” and the ground-truth answer “No.”
>
> We suspect that the main source of confusion may stem from the two subfigures in the lower half of Figure 3. **These subfigures are intended to illustrate why using direct sampling as the decoding strategy can lead to an unfair improvement in model prediction accuracy when the Adaptive Plausibility Constraint is applied.**
>
> - As shown in the bottom-left subfigure, when using the direct sampling strategy under normal conditions, the model predicts the correct answer “No” with a probability of 91.2%, but still produces the incorrect answer “Yes” with a probability of 8.8%. These values can be found in the bar chart labeled "Probs."
>
> According to Equations (2) and (3), the Adaptive Plausibility Constraint penalizes candidates with relatively low predicted probabilities. This is achieved by setting their logits to negative infinity, as illustrated in the two bar charts labeled “logits,” which causes their softmax probabilities to become zero.
>
> - As shown in the bottom-right subfigure, after applying the Adaptive Plausibility Constraint, the logits value for the incorrect answer “Yes” is set to negative infinity, causing its predicted probability to drop to zero. As a result, even though the decoding strategy remains direct sampling, the probability of selecting the correct answer “No” increases from 91.2% to 100% (see the bar charts labeled “Probs” for comparison). As demonstrated in our conclusion, this is equivalent to a degeneration of the direct sampling strategy into greedy search.
>
> **Thank you for pointing out the lack of clarity in the presentation of this figure. We will revise the figure to include clearer and more explicit annotations, and we will also expand the accompanying textual explanation to ensure the figure is easier to interpret.**
>
> ---
>
> # *W2: The main conclusion remains ambiguous. Could the authors clarify the intended implication and its broader significance?*
>
> ## *1. Clarification of Ambiguities in Our Conclusion*
>
> Here, “enhance performance” refers specifically to the improvement in prediction accuracy achieved by methods such as VCD on discriminative datasets like POPE and MME.
>
> **However, this performance gain is illusory**. As shown in Section 5 of our paper, even **two dummy approaches**, Forced Distribution Adjustment and Standalone Application of the Adaptive Constraint, **are able to achieve similar numerical improvements** in prediction accuracy on POPE and MME, comparable to those of methods such as VCD. Furthermore, we demonstrate that on other datasets, including CHAIR, NoCaps, and LLaVA-Bench, methods like VCD can even lead to a decline in prediction accuracy.
>
> This clearly indicates that the numerical performance gains achieved by methods such as VCD primarily result from incidental output distribution adjustments and potentially unfair decoding strategies, as analyzed in Section 4, rather than from genuinely mitigating hallucinations.
>
> ## *2. The Broader Significance of Our Conclusion*
>
> To highlight the significance of this conclusion, **we would first like to provide a brief overview of the current research landscape on hallucination mitigation methods based on the idea of contrastive decoding.**
>
> **Over the past year, more than 30 papers on hallucination mitigation methods based on contrastive decoding have been published at top-tier conferences such as CVPR, ECCV, NeurIPS, and ICLR.** These works generally follow a similar research paradigm: a novel contrastive decoding-based method is proposed, evaluated on benchmark datasets such as POPE and MME, and reported to achieve higher prediction accuracy, thereby claiming superiority in hallucination mitigation.
>
> **However, as we have analyzed and demonstrated in this paper, these proposed methods do not truly mitigate hallucinations.** Instead, their reported improvements stem from coincidental adjustments to output distributions on limited datasets and the potential use of inconsistent decoding strategies, leading to misleading performance gains.
>
> **The number of papers in this area continues to grow, with many researchers investing significant time and effort into what may ultimately be a misguided direction.** Through this work, we aim to correct the current flawed research paradigm, and possibly the underlying research trajectory itself, in order to help the community avoid unnecessary expenditure of resources and effort.
>
> ---
>
> # *Q1:  In lines 170–174, the observation that MLLMs tend to generate “NO” is discussed. Is this tendency specific to the POPE benchmark, or does it generalize to other datasets and evaluation protocols?*
>
> We address this question in two parts:
> - For discriminative tasks (e.g., binary Yes/No questions), the same trend persists not only on the POPE dataset, but also across other datasets.
> - For open-ended generation tasks, the trend may manifest in different forms, but the core conclusion, that contrastive decoding fails to mitigate hallucinations, consistently holds.
>
> ## *1. Discriminative Tasks (Binary Yes/No Answers)*
>
> **Across all discriminative tasks, applying methods such as VCD consistently leads to an output distribution that is biased toward the answer “YES.”** To further support this point, we present the change in the proportion of “YES” responses on the MME dataset. As shown in the table below, the application of VCD significantly increases the frequency of “YES” answers.
>
> |   **Method**  | **MME-Count** | **MME-Color** | **MME-Existence** | **MME-Position** |
> |:----:|:----:|:----:|:----:|:----:|
> | Vanilla | 61.67% | 56.67% | 50.00% | 66.67%  |
> | VCD  | 71.67% | 63.33% | 53.33%  | 73.33%  |
> | Difference  | +10.00% | +6.66%  | +3.33% | +6.66% |
>
> ## *2. Open-Ended Generation Tasks (CHAIR, NoCaps, LLaVA-Bench, etc.)*
>
> **In our paper, we extend the observation of output bias, specifically, the tendency to generate “YES”, to open-ended generation tasks.** Unlike binary tasks where responses are limited to “YES” or “NO,” open-ended tasks allow for more flexible and diverse outputs. As a result, the shift in output distribution cannot be characterized merely by an increased frequency of “YES” responses.
> **In Appendix C,** we provide a qualitative analysis of several examples from open-ended question answering tasks, revealing a more fundamental impact of contrastive decoding methods:
>
> **The contrastive examples do not expose or guide the model to hallucinations; instead, they force the model to make predictions based on incomplete visual inputs.** As a result, the model may randomly penalize highly relevant candidate answers, depending on how the incomplete input is interpreted, which offers no real benefit to predictive accuracy. In some cases, this may even lead to decreased accuracy when the contrastive output happens to align with the correct answer.
>
> **This conclusion is also supported by our experimental results.** As shown in Table 4, applying VCD to captioning tasks such as CHAIR and NoCaps, as well as to open-ended QA tasks like LLaVA-Bench, leads to a decrease in prediction accuracy.
>
> **In summary, the phenomena we reveal in this paper span multiple tasks and datasets.** We chose to focus our in-depth analysis on the POPE dataset because methods such as VCD have been primarily validated on this benchmark. By doing so, we aim to provide a point-by-point response to these prior works, thereby making our argument more direct and comprehensive.
>
> ---
>
> # *Final Acknowledgment*
>
> **We sincerely thank the reviewers once again for their time, effort, and valuable feedback. We hope our responses have resolved the concerns raised, and we are happy to provide additional clarification if needed. We truly appreciate your thoughtful review, and we wish you all the best in your future research endeavors.**

---

> > ### Comment · Reviewer_RnVz · 2025-08-05
> >
> > Thanks for your response and I would maintain my evaluation in light of your rebuttal and discussion with the other reviewers.

---

> > > ### Author Response · Authors · 2025-08-05
> > >
> > > Thank you for your response and for taking the time to consider both our rebuttal and the broader reviewer discussion. We understand and respect your decision to maintain your original evaluation.
> > >
> > > At the same time, we would like to kindly confirm whether all of your concerns have been fully addressed. If there are any remaining issues or clarifications needed, we would be more than happy to provide further responses.
> > >
> > > Thank you again for your time and thoughtful review. Wishing you all the best in your future research endeavors.

---

### Official Review · Reviewer_z19D · 2025-07-02

**Clarity:** 3
**Significance:** 2
**Originality:** 3
**Rating:** 4
**Confidence:** 4

**Summary:**

The authors show that contrastive decoding methods that show improvement on the POPE dataset are actually due to spurious reasons, namely 1. the imbalance of yes's to no's, and 2. the collapse to greedy search due to the the adaptive plausibility constraint.

**Questions:**

1. Do these findings generalize beyond the POPE dataset? What about datasets like TruthfulQA or FactScore that have non-binary outputs?
2. The model evaluations are quite limited -- the paper only evaluates three models in 7B and 13B. Are larger models beyond 13B still susceptible to this type of bias? It would be useful to show a trend -- does this class imbalance issue worsen or improve with increased model size?
3. Please discuss metrics to address shortcomings in binary accuracy evaluation, like per-class precision/recall and F1. Performance should be reported in terms of these metrics as well for clarity.
4. Please include examples illustrating your finding, i.e., questions that were previously true negatives that got flipped to false positives due to this bias.
5. The discussion should include proposals for clear alternatives and solutions for this issue.
6. Please clarify the COCO and GQA subsets of the POPE dataset; the authors seem to assume reader knowledge of the dataset which may not always be the case.
7. Equation 3 has p_0 on the left; should this be p_\theta?
8. Tables 5 and 6 use Sample† and Sample* interchangeably; please keep one notation.

**Ethical Concerns:**

["NO or VERY MINOR ethics concerns only"]

**Final Justification:**

The authors provide a clear problem with sufficient evidence to back up their claims. I have updated the score based on their provided feedback.

**Limitations:**

yes

**Quality:**

3

**Strengths And Weaknesses:**

Strengths: The paper notes a significant limitation in the evaluations of VLLMs on the POPE dataset. This is an important finding for past and future works that cite the POPE dataset.

Weaknesses:
1. It's unclear that showing a limitation of the POPE dataset is sufficient for acceptance as a conference paper. The authors should further provide background on the impact of the POPE dataset in the broader research field of hallucination detection. As it stands, the paper is not strong enough in my opinion as as a standalone work.
2. The authors should evaluate other hallucination benchmarks that move beyond binary QA (i.e., free-text, caption-based outputs). One experiment that would strengthen the paper would be to demonstrate this effect on non-binary QA benchmarks as well. Class imbalance is a problem with all binary QA datasets (not unique to POPE), and there are metrics for addressing this.

---

> ### Author Rebuttal · Authors · 2025-07-31
>
> # *Initial Acknowledgment*
>
> **We are grateful to the reviewers for their valuable and encouraging feedback. The insightful suggestions and comments helped us identify areas for clarification and further improvement. In the following, we address each of the reviewers’ concerns point by point.**
>
> ---
>
> # *W1: It's unclear that showing a limitation of the POPE dataset is sufficient for acceptance as a conference paper.*
>
> **We believe there may be a significant misunderstanding here. In fact, the main focus of our paper is not to highlight a limitation of  POPE dataset, but rather to demonstrate that hallucination mitigation methods based on the idea of contrastive decoding are fundamentally ineffective**, despite the fact that over a hundred such methods have been proposed in the past year. **Please allow me to clarify the central arguments of our paper in detail below.**
>
> **First, allow us to briefly introduce the current state of research on hallucination mitigation methods based on the concept of contrastive decoding.** Over the past year, more than 30 studies on contrastive decoding-based hallucination mitigation have been published at top-tier conferences such as CVPR, NeurIPS, and ICLR. These works generally follow a similar research paradigm: they propose a novel hallucination mitigation strategy grounded in contrastive decoding, evaluate its performance on benchmarks such as POPE and MME, and report improved prediction accuracy—thereby demonstrating the effectiveness of their proposed method.
>
> **The purpose of this paper is to demonstrate that this class of methods (such as VCD, ICD, SID, and many others of a similar kind) is fundamentally ineffective.**
>
> **We first demonstrate that the accuracy improvements reported by methods such as VCD on POPE and MME are in fact illusory.** In Section 4 of the paper, we analyze how the apparent performance gains on POPE and MME primarily stem from two misleading factors:
> - crude, unidirectional adjustments to the model’s output distribution
> - the adaptive plausibility constraint, which effectively reduces the sampling strategy to a greedy search.
>
> Furthermore, in Section 5, we introduce two dummy methods, PBA and OLM, and show that they can easily achieve similar improvements on POPE and MME. This further supports our claim that the observed accuracy gains of methods like VCD do not result from a genuine reduction in hallucination.
>
> **In addition, we broaden the scope of our experiments to cover a wider range of tasks, and find that methods such as VCD actually lead to a degradation in performance.** Specifically, we evaluate these methods on image captioning tasks (CHAIR, NoCaps) and an open-ended question answering task (LLaVA-Bench), as reported in Table 4 and Table 6. Our results indicate that in open-ended generation scenarios, these methods not only fail to help, but can even reduce accuracy.
>
> **Building on the above two points, we provide a detailed demonstration that methods such as VCD are entirely ineffective at mitigating hallucinations.** Nevertheless, research in this area continues to proliferate, with numerous researchers investing significant time and effort into a fundamentally flawed direction. With this paper, we aim to correct the prevailing, yet misguided, research paradigm—and potentially the research trajectory itself—**in order to help the community avoid unnecessary expenditure of resources and effort.**
>
> ---
>
> # *W2&Q1: The authors should evaluate other hallucination benchmarks that move beyond binary QA. What about datasets like TruthfulQA or FactScore that have non-binary outputs?*
>
> **Importantly, our experiments are not limited to discriminative tasks such as POPE. As shown in Tables 4 and 6 of the paper, we have extended our evaluation to include open-ended generation tasks as well.** These tasks span a diverse set of benchmarks:
>
> - Discriminative tasks: POPE, MME
> - Captioning tasks: CHAIR, NoCaps
> - Open-ended QA task: LLaVA-Bench
>
> **Additionally, Appendix C presents a qualitative analysis demonstrating that the core mechanism of contrastive example processing is prediction based on partial image inputs**, thereby illustrating how methods such as VCD impose coarse modifications on output distributions in open-ended generative tasks.
>
> **We appreciate your interest in the TruthfulQA and FactScore datasets—both are excellent benchmarks. However, they are not suitable for evaluating methods such as VCD, ICD, and SID,** as these approaches require image inputs. Since TruthfulQA and FactScore are designed for purely text-based tasks, they cannot accommodate image-conditioned methods like VCD.
>
> ---
>
> # *Q2: Are larger models beyond 13B still susceptible to this type of bias?*
>
> **We are somewhat puzzled, as the forced modification of the output distribution originates from the nature of methods like VCD themselves, and is independent of the underlying model size, as analyzed in Section 4 of our paper.** Are you suggesting that such methods might not affect the output distribution when applied to larger-scale models? In any case, we have demonstrated the biased output distribution (YES%) on the LLaVA-NeXT-34B model, as shown in the table below.
>
> | **Method** | **Random**   | **Popular**  | **Adversarial** |
> |:----------:|:----------:|:---------:|:-----------:|
> | Greedy | 41.4 ↑0.0 | 42.5 ↑0.0 | 45.0 ↑0.0   |
> | VCD | 48.5 ↑7.1| 51.3 ↑8.8 |   55.2 ↑10.2   |
>
> By comparing Table 3 in the paper with the table above, we observe that larger-scale models yield improved initial prediction accuracy, partially mitigating the initial output bias. **However, methods such as VCD continue to enforce output distribution modifications (with a bias toward "Yes") with consistent intensity, showing no reduction in the degree of adjustment.**
>
> ---
>
> # *Q3: Please discuss metrics to address shortcomings in binary accuracy evaluation, like per-class precision/recall and F1.*
>
> Thank you for the suggestion. Due to space constraints, we only report precision, recall, and F1-score on the POPE-COCO-Random dataset in the table below. Full results on the complete POPE dataset will be added in the appendix.
>
> | **Method** | **Precision**       | **Recall**          | **F1-Score**        |
> |:----------:|:------------------:|:-------------------:|:-------------------:|
> | Greedy     | 97.3% ↑0.0%    | 76.2% ↑0.0%      | 85.5%  ↑0.0%   |
> | VCD        | 91.5% ↓5.8%       | 85.0% ↑8.8%        | 88.1% ↑2.6%     |
>
> The table shows a notable drop in precision and a clear increase in recall. This is consistent with the fact that the model tends to predict "NO" more often on COCO-Random, while methods like VCD shift the output distribution toward "YES."
>
> ---
>
> # *Q4: Please include examples illustrating your finding, i.e., questions that were previously true negatives that got flipped to false positives due to this bias.*
>
> Thank you for your suggestion. As external links are not permitted in the rebuttal, we can only provide the textual portion of the example (with the image input omitted), as shown below. The complete example, including the image, will be included in the appendix of the revised paper.
>
> - {"question_id": 16, "prompt": "Is there a motorcycle in the image?", "vanilla-text": "No", "vcd-text": "Yes", "image": "COCO_val2014_000000429109.jpg"}
> - {"question_id": 82, "prompt": "Is there a bowl in the image?", vanilla-text": "No", "vcd-text": "Yes", "image": "COCO_val2014_000000569839.jpg"}
>
> ---
>
> # *Q5: The discussion should include proposals for clear alternatives and solutions for this issue.*
>
> Hallucination mitigation is a broad and active research area. **This paper solely aims to demonstrate that hallucination mitigation methods based on contrastive decoding are ineffective.** However, there exist many other effective approaches to address hallucination. For example:
>
> - Reinforcement learning-based fine-tuning strategies for hallucination mitigation. [1]
>
> - Test-time intervention techniques designed around latent knowledge space. [2]
>
> ---
>
> # *Q6: Please clarify the COCO and GQA subsets of the POPE dataset.*
>
> **Thank you for your valuable suggestion. We would like to briefly provide some background information.** POPE originally refers to a polling-based query method proposed by the authors. In the original POPE paper, this method was applied to images from three datasets—COCO, AOKVQA [3], and GQA [4]—which together constitute the widely recognized POPE benchmark. Accordingly, COCO, AOKVQA, and GQA refer specifically to the image sources utilized. A detailed introduction to POPE will be included in the appendix.
>
> ---
>
> # *Q7&Q8: Equation 3 has $p_0$ on the left; should this be $p_\theta$? Tables 5 and 6 use Sample† and Sample\* interchangeably; please keep one notation.*
>
> Yes, you are absolutely right. In Equation (3), it should indeed be $p_\theta$, and the term "Sample†" should be used consistently in both Table 5 and Table 6. Thank you for pointing out these issues. We apologize for our oversight and will make the necessary corrections.
>
> ---
>
> # *Final Acknowledgment*
>
> **We sincerely thank the reviewers once again for their time, effort, and valuable feedback. We hope our responses have resolved the concerns raised, and we are happy to provide additional clarification if needed. We truly appreciate your thoughtful review, and we wish you all the best in your future research endeavors.**
>
> ---
>
> [1] Yang, Zhihe, et al. "Mitigating hallucinations in large vision-language models via dpo: On-policy data hold the key." CVPR 2025.
> [2] Liu, Sheng, et al. "Reducing hallucinations in vision-language models via latent space steering." arXiv preprint arXiv:2410.15778 (2024).
> [3] Schwenk, Dustin, et al. "A-okvqa: A benchmark for visual question answering using world knowledge." ECCV 2022.
> [4] Hudson, Drew A., and Christopher D. Manning. "Gqa: A new dataset for real-world visual reasoning and compositional question answering." CVPR 2019.

---

> > ### Comment · Reviewer_z19D · 2025-08-04
> >
> > Thank you, you have adequately addressed my concerns.

---

> > > ### Author Response · Authors · 2025-08-05
> > >
> > > Thank you very much for your feedback. We’re glad to hear that our responses have adequately addressed your concerns. We will ensure that all the relevant clarifications and improvements introduced during the rebuttal are carefully incorporated into the final version of the paper.
> > >
> > > If you are satisfied with the outcome of the rebuttal, we would truly appreciate it if you could consider raising your score.
> > >
> > > Thank you again for your valuable time and constructive insights. We wish you all the best in your future research endeavors.

---

### Official Review · Reviewer_q7te · 2025-07-03

**Clarity:** 3
**Significance:** 3
**Originality:** 3
**Rating:** 4
**Confidence:** 4

**Summary:**

This paper questions the effectiveness of contrastive decoding in reducing object hallucinations in multimodal LLMs. It shows that the observed improvements on the POPE benchmark are primarily due to biased output adjustments and the adaptive plausibility constraint rather than genuine mitigation of hallucinations.

**Questions:**

* How can future benchmarks be better designed to measure hallucination accurately?
* Can contrastive decoding be restructured to correct both false positives and false negatives?
* How would these findings extend to open-ended generation tasks beyond binary QA?

**Ethical Concerns:**

["NO or VERY MINOR ethics concerns only"]

**Limitations:**

Yes

**Paper Formatting Concerns:**

Figure 1 and Figure 2 contain mixed font sizes, uneven alignment, and unclear legends.

**Quality:**

3

**Strengths And Weaknesses:**

Strengths:

* This paper provides a thorough analysis of contrastive decoding, identifying its key limitations through empirical and theoretical evidence.
* This paper introduces well-designed baselines that isolate and reveal the misleading factors behind observed performance gains.
* This paper conducts comprehensive experiments across models and tasks, strengthening the validity of its conclusions.
* This paper offers practical insights into evaluation practices and future directions in hallucination mitigation research.

Weaknesses:
* This paper relies heavily on the POPE benchmark, raising concerns about generalizability to more complex or real-world scenarios.
* This paper does not explore or compare with other hallucination mitigation techniques beyond contrastive decoding.
* This paper assumes a consistent response bias pattern (“Yes”/“No”) across datasets, which may not always hold in broader contexts.

---

> ### Author Rebuttal · Authors · 2025-07-28
>
> # *Initial Acknowledgment*
>
> **We are grateful to the reviewers for their valuable and encouraging feedback. The insightful suggestions and comments helped us identify areas for clarification and further improvement. In the following, we address each of the reviewers’ concerns point by point.**
>
> ---
>
> # *W1: This paper relies heavily on the POPE benchmark, raising concerns about generalizability to more complex or real-world scenarios.*
>
> **We appreciate your concern. Nevertheless, our study extends beyond the scope of POPE.** As shown in Tables 4 and 6 of the paper, we have extended our analysis to more realistic scenarios, encompassing a broader range of tasks, including discriminative tasks, captioning tasks, and open-ended QA tasks**. Specifically:
>
> - Discriminative tasks: POPE, MME
> - Captioning tasks: CHAIR, NoCaps
> - Open-ended QA tasks: LLaVA-Bench
>
> Thus, we have in fact demonstrated the ineffectiveness of contrastive decoding-based hallucination mitigation methods across a diverse set of tasks.
>
> That said, as you rightly pointed out, we did devote significant space in our paper to analyzing the performance of hallucination mitigation methods specifically on the POPE benchmark. This is because the papers proposing methods such as VCD primarily used POPE to validate their effectiveness. Our aim is to demonstrate the ineffectiveness of these methods in a point-by-point manner, thereby strengthening the persuasiveness of our paper.
>
> ---
>
> # *W2: This paper does not explore or compare with other hallucination mitigation techniques beyond contrastive decoding.*
>
> Indeed, you are correct. As you have noted, the core objective of our paper is to verify the ineffectiveness of hallucination mitigation methods designed around contrastive decoding, as reflected in our title: "Why Contrastive Decoding Fails to Mitigate Object Hallucinations in MLLMs."
>
> **Hallucination mitigation is a broad and evolving research field, and we do not intend to suggest that all mitigation methods are ineffective, that would clearly be an inaccurate conclusion**. For instance, approaches based on reinforcement learning fine-tuning have shown promising effectiveness to date.
>
> **However, it is important to emphasize that demonstrating the ineffectiveness of contrastive decoding alone in mitigating hallucinations holds substantial value for the research community.** Hallucination mitigation methods based on contrastive decoding have become a widely adopted and influential line of work. Over the past year, more than 30 papers following this paradigm have been published at top-tier conferences, including CVPR, ICLR, NeurIPS, AAAI, and ICML. By revealing the fundamental limitations of this research paradigm, our findings can help the community avoid unnecessary expenditure of resources and effort, thereby redirecting attention toward more promising directions.
>
> ---
>
> # *W3: This paper assumes a consistent response bias pattern (“Yes”/“No”) across datasets, which may not always hold in broader contexts.*
>
> **We believe there may be a misunderstanding. We do not assume that every dataset exhibits a consistent bias pattern.** In fact, aside from POPE, the other evaluation datasets used in our paper—MME, CHAIR, NoCaps ,and LLaVA-Bench—do not display any clear or systematic bias of this kind.
>
> **However, on datasets that do not exhibit such bias pattern, applying hallucination mitigation methods like VCD leads to a clear drop in prediction accuracy (see Table 4 in our paper).** This further supports our conclusion: the accuracy improvements observed with VCD on the POPE dataset are incidental and primarily driven by the unique bias patterns present in MLLM outputs on POPE. When evaluated across a broader range of tasks and datasets, **the experimental results provide even stronger evidence that methods like VCD are, in fact, ineffective.**
>
> ---
>
> # *Q1: How can future benchmarks be better designed to measure hallucination accurately?*
>
> Thank you for your excellent question. When it comes to object hallucination, the primary focus of our paper, we believe that a well-designed hallucination benchmark should satisfy two key criteria: **question clarity** and **response complexity**.
>
> - **question clarity:** The textual output of MLLMs tends to be more diverse and flexible compared to that of LLMs, primarily because images inherently contain far more information than text. As a result, any textual description of an image inevitably involves a degree of abstraction and selective emphasis. For example, a caption might omit a small cup located in the upper-left corner of the image. However, if the model is explicitly asked whether there is a cup in the upper-left corner, it may respond “yes.” This illustrates why captioning tasks are often unsuitable for evaluating hallucinations in MLLMs: the omission of certain objects may reflect linguistic conventions or summarization strategies rather than true hallucinations. **Therefore, we argue that object hallucination evaluation in MLLMs requires explicit and well-defined questions.**
>
> - **Response Complexity**: Datasets like POPE are indeed designed with the goal of asking clearly defined questions. However, their binary (yes/no) format results in minimal information in the response, which makes it difficult to assess whether the model truly understands the concept of the object, has accurately located it within the image, or has simply guessed correctly by chance. This ambiguity complicates the evaluation of hallucinations. Therefore, we believe that a series of short, content-rich questions would be a more effective testing approach. For instance: "What is the object in the center of the image?", "What color is it?", or "What object is to its right?" **Compared to binary classification tasks, such short-form Q&A prompts elicit more detailed answers, making it easier to identify hallucinations in object understanding.**
>
> ---
>
> # *Q2: Can contrastive decoding be restructured to correct both false positives and false negatives?*
>
> First, we can confidently state that current hallucination mitigation methods based on the principle of contrastive decoding are incapable of simultaneously correcting both false positives and false negatives. **Looking ahead, is it possible to redesign contrastive decoding to address both issues at once? We remain skeptical.**
>
> **The core issue lies in a key assumption underlying contrastive decoding-based hallucination mitigation: that object hallucinations arise primarily from overly strong language priors**. As a result, most contrastive decoding methods are designed to weaken the visual input in order to expose these language priors, thereby suppressing them during generation.
>
> However, our experiments challenge this assumption in two ways:
>
> - When the visual input is weakened, the model does not exhibit the expected hallucinations. Instead, it generates outputs based solely on the limited visual features that remain (**see Appendix C for details**).
> - The outputs of the contrastive samples, which are intended to reflect language priors, often do not align with the hallucinated predictions. This suggests that multimodal object hallucination may not necessarily stem from strong language priors.
>
> Taken together, these findings suggest that we are still far from developing a contrastive decoding strategy capable of simultaneously correcting both types of errors.
>
> ---
>
> # *Q3: How would these findings extend to open-ended generation tasks beyond binary QA?*
>
> **Indeed, our paper extends the analysis to open-ended generation tasks.** Although the flexible nature of outputs in such tasks makes it difficult to perform the kind of direct quantitative evaluation used on POPE, **we provide a qualitative analysis in Appendix C**. Through several open-ended QA examples, we show that the model’s handling of contrastive samples is largely driven by partial visual inputs, rather than the generation of the hypothesized hallucinations.
>
> **Furthermore, as shown in Tables 4 and 6 of our paper, we evaluate the performance of methods like VCD on captioning tasks (CHAIR, NoCaps) and open-ended QA tasks (LLaVA-Bench)**. We find that on these datasets, which do not exhibit clear output biases, **these methods actually lead to a decline in prediction accuracy.**
>
> To summarize our experimental findings:
>
> - On the POPE dataset, methods like VCD lead to improved prediction accuracy. **However, our analysis shows that this improvement stems from shifts in the output distribution, rather than genuine mitigation of hallucinations.** Notably, the original papers proposing these methods primarily reported results on POPE, which has led many to believe that such methods are effective.
>
> - **On other datasets, these methods even result in decreased prediction accuracy**, which further demonstrates that contrastive decoding strategies fail to effectively mitigate hallucinations.
>
> As the above results demonstrate, hallucination mitigation methods based on the contrastive decoding paradigm are **ineffective across all types of hallucination evaluation tasks.**
>
> ---
>
> # *Final Acknowledgment*
>
> **We sincerely thank the reviewers once again for their time, effort, and valuable feedback. We hope our responses have resolved the concerns raised, and we are happy to provide additional clarification if needed. We truly appreciate your thoughtful review, and we wish you all the best in your future research endeavors.**

---

> > ### Author Response · Authors · 2025-08-06
> >
> > Dear Reviewer,
> >
> > I hope this message finds you well. As the discussion period is nearing its end with less than three days remaining, I wanted to ensure we have addressed all your concerns satisfactorily. If there are any additional points or feedback you'd like us to consider, please let us know. Your insights are invaluable to us, and we’re eager to address any remaining issues to improve our work.
> >
> > Thank you for your time and effort in reviewing our paper.

---

### Author Response · Authors · 2025-08-09
**Final General Response**

**We are grateful for the opportunity to engage in constructive discussions with the reviewers during the review period.** The exchange of ideas has been invaluable in refining both the presentation and scope of our work, and we are encouraged by the recognition of its key contributions and strengths.

In particular, we appreciate the acknowledgment of our empirical studies, which critically challenge the contrastive decoding paradigm in MLLM hallucination mitigation and provide evidence that its reported performance gains may be misleading (**Reviewer #1-q7te, Reviewer #3-RnVz, Reviewer #4-bPyU**). We are also grateful for the recognition that our experimental setup is well designed—featuring multiple models, diverse benchmarks, and carefully crafted control techniques to reveal the paradigm’s core weakness (**Reviewer #1-q7te, Reviewer #4-bPyU**). Furthermore, we appreciate the acknowledgment that our analysis of why these methods achieve higher scores—by balancing answers or altering the search strategy—is both clear and persuasive (**Reviewer #3-RnVz**). Finally, we are pleased that our analyses were considered valuable in advancing the community’s understanding of the true causes of hallucination reduction (**Reviewer #1-q7te, Reviewer #3-RnVz, Reviewer #4-bPyU**).

---

During the discussion period, we received valuable suggestions from the reviewers. We have carefully addressed each comment and believe that we have satisfactorily responded to the majority of their concerns. We will incorporate the suggested experiments, additional discussions, and relevant updates to further strengthen our work. Below, we summarize the issues addressed in our rebuttal, along with the reviewers’ acknowledgment of our responses.

---

***Reviewer #1-q7te’s main question focuses on the generalizability of our conclusions regarding the hallucination mitigation strategy to open-ended generation tasks beyond those evaluated on the POPE dataset.***

In our rebuttal, we noted that Section 5 of our paper already extends the experimental scope to the image captioning task (NoCaps CHAIR) and the open-ended QA task (LLaVA-Wild). In addition, Appendix C offers a more detailed qualitative analysis of the results on the open-ended QA task. Although the reviewer did not participate in the discussion phase, we believe our response adequately addresses their concerns.

---

***Reviewer #2-z19D placed particular emphasis on our discussion of the POPE dataset’s limitations in the initial review, which appeared to influence their assessment of our paper’s overall contribution.***

In our rebuttal, we clarified that the central aim of our paper is to demonstrate that hallucination mitigation methods based on the idea of contrastive decoding—a research direction that has been highly active over the past year with more than 30 papers presented at top-tier conferences—are fundamentally ineffective. **The reviewer ultimately acknowledged our response, stating, “Thank you, you have adequately addressed my concerns.”**

---

***Reviewer #3-RnVz pointed out certain ambiguities in our notation and phrasing, and also raised concerns about whether our conclusions on the POPE dataset could be generalized to broader domains.***

In our rebuttal, we clarified the notation and certain expressions. We also explained that, as presented in Section 5 and Appendix C, our conclusions have already been extended to the open-ended QA and image captioning tasks. The reviewer acknowledged our response and noted that they would maintain a positive score, commenting: “Thanks for your response, and I will maintain my evaluation in light of your rebuttal and discussion with the other reviewers.”

---

***Reviewer #4-bPyU expressed interest in seeing experiments conducted on a broader range of base models, hallucination mitigation methods, and datasets.***

In our rebuttal, we provided these additional results. The reviewer appreciated the expanded experiments and subsequently raised their score, stating: “Thanks for the rebuttal and extensive added experimentations. I will raise the score.”

---

***Finally, we sincerely thank the reviewers for their constructive feedback, which has significantly improved both the clarity and substance of our work. We are grateful for the opportunity to engage in this constructive exchange and to further strengthen our research through the process. We wish the reviewers and area chairs continued success in their future research endeavors.***

---

### Decision · Program_Chairs · 2025-09-17

**Decision:**

Accept (poster)

**Comment:**

This paper provides a careful and insightful analysis of contrastive decoding for hallucination mitigation in VLLMs. It identifies key limitations through both empirical and theoretical evidence, supported by well-designed baselines that isolate misleading factors behind observed performance gains. The experiments are comprehensive, spanning multiple models and tasks, and the findings offer practical insights into fair evaluation practices and future directions for hallucination research.

The work highlights an important flaw in the evaluation of VLLMs on the POPE dataset and convincingly shows that apparent improvements from contrastive decoding often arise from unbalanced answers or altered search strategies rather than genuine mitigation. This critique is clear, well-argued, and of strong relevance to the community, especially given the real-world safety implications.

Overall, this is a well-executed, timely, and impactful contribution. Most reviewers recommend acceptance.